# Allosteric cooperation in β-lactam binding to a non-classical transpeptidase

Nazia Ahmad[1†], Sanmati Dugad[2†], Varsha Chauhan[2], Shubbir Ahmed[3], Kunal Sharma[1], Sangita Kachhap[4], Rana Zaidi[1], William R Bishai[2], Gyanu Lamichhane[2]*, Pankaj Kumar[1,2]*

[1]Department of Biochemistry, School of Chemical and Life Sciences, Jamia Hamdard University, Delhi, India; [2]Department of Infectious Diseases, Centre for Tuberculosis Research, Johns Hopkins University, Baltimore, United States; [3]Translational Health Science and Technology Institute, NCR Biotech Science Cluster, Faridabad, India; [4]Jerzy Haber Institute of Catalysis and Surface Chemistry, Polish Academy of Sciences, Niezapominajek, Poland

**Abstract** L,D-transpeptidase function predominates in atypical 3 → 3 transpeptide networking of peptidoglycan (PG) layer *in Mycobacterium tuberculosis*. Prior studies of L,D-transpeptidases have identified only the catalytic site that binds to peptide moiety of the PG substrate or β-lactam antibiotics. This insight was leveraged to develop mechanism of its activity and inhibition by β-lactams. Here, we report identification of an allosteric site at a distance of 21 Å from the catalytic site that binds the sugar moiety of PG substrates (hereafter referred to as the S-pocket). This site also binds a second β-lactam molecule and influences binding at the catalytic site. We provide evidence that two β-lactam molecules bind co-operatively to this enzyme, one non-covalently at the S-pocket and one covalently at the catalytic site. This dual β-lactam-binding phenomenon is previously unknown and is an observation that may offer novel approaches for the structure-based design of new drugs against *M. tuberculosis*.

*For correspondence:
gyanu@jhu.edu (GL);
pkumar10@jhmi.edu (PK)

[†]These authors contributed equally to this work

Competing interest: The authors declare that no competing interests exist.

## Editor's evaluation

This manuscript reports high-resolution crystallographic structures of the L,D-transpeptidase from *Mycobacterium tuberculosis*, which was obtained with ligands (a sugar molecule and a β-lactam). A surprising finding is that the enzyme contains a ligand-binding site located greater than 20 Å away from the catalytic site. The authors used mutagenesis studies and computational analyses to support an allosteric role for the new ligand site (S-pocket). This site could potentially permit the inhibition of L,D-transpeptidases.

## Introduction

Tuberculosis (TB), is a major threat to global health as it claims more human lives than any other bacterial infection (*Chakaya et al., 2021*). The emergence of multi- and extensively drug-resistant strains of *Mycobacterium tuberculosis (M.tb)*, the bacterium that causes TB, has further limited our capability to fight the disease. One major factor contributing to the emergence of drug-resistant TB is poor compliance with prolonged treatment regimens. While combinatorial drug therapy kills the majority of *M.tb* bacilli, a subset of the bacterial population, defined as 'persisters', tolerate TB drugs. This persister subset requires prolonged treatment for sterilization to occur (*Gideon and Flynn, 2011*; *Lillebaek et al., 2002*; *Peddireddy et al., 2017*; *Wayne and Hayes, 1996*). Mechanisms of persistence in *M.tb are* likely multifactorial, involving cell wall peptidoglycan (PG) remodelling, transporters or efflux

pumps, and alternative energy sources (*Keren et al., 2011*; *Zhang et al., 2012*). Molecular understanding of these pathways may facilitate discovery of new therapeutics to overcome the current challenges posed by *M.tb* persistence and drug resistance.

PG is an essential component of the bacterial cell wall and constitutes the exoskeleton of bacterial cells. PG consist of long glycan chains composed of two different sugars *N*-acetyl muramic acid and *N*-acetylglucosamine that are crossed linked via stem peptide chains. The PG composition of *M.tb* in slowly replicating states is likely to be distinct from the one during active growth (*Gupta et al., 2010*; *Schoonmaker et al., 2014*; *Wietzerbin et al., 1974*). In particular, a high percentage of peptide crosslinks in *M.tb* join the third amino acids (3–3 linkages) of the adjacent stem peptides instead of the classical 4–3 linkages, and these linkages are formed by transpeptidases (*Tolufashe et al., 2020*). The 4–3 linkages, which were historically considered to predominate throughout bacterial growth and senescence, are generated by a well-known enzyme class, namely the D,D-transpeptidases (also known as penicillin-binding proteins) (*Tolufashe et al., 2020*). The 3–3 linkages are generated by the more recently discovered enzyme class, the L,D-transpeptidases (*Mainardi et al., 2005*).

Among the five L,D-transpeptidase paralogs of *M.tb*, Ldt$_{Mt2}$ plays an important role since an *M.tb* strain lacking the gene encoding this enzyme exhibits attenuation in virulence and exhibits enhanced susceptibility to β-lactam antibiotics (*Bianchet et al., 2017*; *Brammer Basta et al., 2015*; *Dubée et al., 2012*; *Gupta et al., 2010*; *Libreros-Zúñiga et al., 2019*; *Sanders et al., 2014*; *Schoonmaker et al., 2014*). Additional reports have demonstrated altered and attenuated cellular physiology of *M.tb* in association with the loss of function of Ldt$_{Mt1}$ (*Schoonmaker et al., 2014*) and Ldt$_{Mt5}$ (*Brammer Basta et al., 2015*). The requirement of L,D-transpeptidases for virulence has suggested that they may comprise valuable drug targets, and indeed these enzymes are inhibited by the carbapenem subclass of β-lactam drugs (*Mainardi et al., 2005*). Recent work further suggests the efficacy of these carbapenems against both dividing and non-dividing mycobacteria (*Hugonnet et al., 2009*). To further evaluate the druggability of the L,D-transpeptidases class, several independent groups have described the crystal structures of this enzyme class bound to carbapenems such as meropenem, tebipenem, biapenem, and faropenem (*Bianchet et al., 2017*; *Erdemli et al., 2012*; *Kim et al., 2013*; *Li et al., 2013*; *Steiner et al., 2017*). The biochemical mechanisms and kinetics of inhibition of L,D-transpeptidases by β-lactams have been documented (*Cordillot et al., 2013*), and new experimental β-lactams that target *M.tb* L,D-transpeptidases have recently been described (*Bianchet et al., 2017*; *Kumar et al., 2017*; *Martelli et al., 2021*).

The L,D-transpeptidase class in *M.tb* is comprised of at least four substructural domains including two immunoglobulin-like domains (IgD1 and IgD2), a YkuD domain, and a C-terminal subdomain (CTSD). The YkuD domain is known to play a role in catalytic function and β-lactam binding (*Erdemli et al., 2012*), while the roles of the other domains remain less certain. The YkuD domain has a highly conserved motif HXX14-17[S/T]HGChN containing three residues analogous to the catalytic triad of cysteine proteases: a cysteine, a histidine and a third residue (Cysteine 354, Histidine 336, and Serine 337 in Ldt$_{Mt2}$) to catalyse the transpeptidation reaction (*Erdemli et al., 2012*). This catalytic triad residues resides under a flap formed by a long loop. The flap can open and close to create two cavities (the inner and outer cavities), around a cysteine residue, that are connected by a narrow tunnel (*Fakhar et al., 2017*). It is proposed that these cavities are binding sites for the acyl acceptor and acyl donor tetrapeptide stems (L-alanyl-D-Glutamyl-*meso*-diaminopimelyl-D-alanine) with the donor tetrapeptide binding to the outer cavity and the acceptor tetrapeptide to the inner cavity (*Erdemli et al., 2012*). As the β-lactam class of drugs mimics the tetrapeptide stems of PG, several of the carbapenems have been found to bind both the inner and outer cavities to form covalent linkage with catalytic cysteine residue of the L,D-transpeptidases (*Bianchet et al., 2017*; *Kim et al., 2013*; *Kumar et al., 2017*).

Despite the significance of L,D-transpeptidases in *M.tb* cell physiology and TB disease, the structural and molecular details of how different chemical groups of the nascent PG structure interact with this enzyme class are not sufficiently understood. The disaccharide-tetrapeptide *N*-acetylglucosamine-*N*-acetylmuramic acid-L-alanyl-D-glutamyl-*meso*-diaminopimelyl-D-alanine is the substrate for L,D-transpeptidases (*Cordillot et al., 2013*; *Lavollay et al., 2008*) while the D,D-transpeptidases use the disaccharide-pentapeptide *N*-acetylglucosamine-*N*-acetylmuramic acid-L-alanyl-D-glutamyl-*meso*-diaminopimelyl-D-alanyl-D-alanine as their substrate (*Tolufashe et al., 2020*). Interactions between the peptide subunits of these nascent PG substrates with their relevant enzymes have been described

(*Cordillot et al., 2013*; *Erdemli et al., 2012*; *Fakhar et al., 2017*; *Lavollay et al., 2008*; *Mainardi et al., 2005*), and it is generally assumed that the disaccharide component of the subunit interacts only with transglycosylases (*Fibriansah et al., 2012*; *Mavrici et al., 2014*) and thus are not relevant to the transpeptidases. However, evidence challenging this model is growing. Recent studies with the *Bacillus subtilis* L,D-transpeptidase $Ldt_{Bs}$ have suggested binding of the disaccharide component through a PG recognition domain, LysM, within $Ldt_{Bs}$ (*Schanda et al., 2014*). However, the mechanism of PG recognition may be different in *M.tb* since this LysM domain is absent in its L,D-transpeptidases.

In the current study, we investigate the interaction of PG substrate and β-lactam with L,D-transpeptidase, $Ldt_{Mt2}$. Based on our structural, biophysical, and biochemical data, we identify a new PG disaccharide moiety-binding pocket (named as S-pocket) at a distance of 21 Å from the catalytic site in $Ldt_{Mt2}$. This new site recognizes β-lactams and modulates their binding and hydrolysis in the catalytic site. Our experimental and computational studies identify allosteric communications between catalytic site and S-pocket that has led to the finding that $Ldt_{Mt2}$ can recognize two β-lactams, one non-covalently at the S-pocket and one covalently at catalytic site. The crystallographic studies of $Ldt_{Mt2}$ with an experimental carbapenem drug further reveal the high-resolution details of allosteric alterations that span between the S-pocket and catalytic site during β-lactam binding. Identification of an allosteric cooperativity between the S-pocket and catalytic site also represents a valuable case to investigate recognition of natural substrates prior to 3–3 transpeptide reaction. The study also

**Table 1.** Data collection and refinement statistics.

|  | $Ldt_{Mt2}$–sugar | $Ldt_{Mt2}$–T203 |
|---|---|---|
| Data collection |  |  |
| Wavelength (Å) | 1.0 | 1.0 |
| Resolution (Å) | 29.73–1.58 (1.64–1.58) | 30.0–1.7 (1.73–1.70) |
| Space group | P 1 21 1 | P 1 21 1 |
| Unit cell (Å) | 60.906, 93.981, 75.539, 90, 92.975, 90 | 60.799 94.278 75.707 90.00 93.14 90.00 |
| Unique reflections[a] | 111,390 | 90,418 |
| Multiplicity[a] | 4.3 (4.1) | 5 (5) |
| Completeness[a] | 96.0 (98.6) | 97.4 (95.9) |
| $R_{merge}$[a, b] | 0.048 (0.55) | 0.073 (0.74) |
| Overall $I/\sigma(I)$[a] | 20.37 (1.8) | 23.7 (3.3) |
| Refinement |  |  |
| $R_{work}$ (%)[c] | 0.1662 | 0.1666 |
| $R_{free}$ (%)[d] | 0.1980 | 0.1853 |
| r.m.s.d. |  |  |
| Bonds (Å) | 0.009 | 0.009 |
| Angles (°) | 1.03 | 1.033 |
| Average $B$-factor (Å²) |  |  |
| Protein | 14.9 | 14.2 |
| Active-site ligand | 19.28 | L01 = 25.71, T20 = 34.1 |
| Ramachandaran |  |  |
| Favoured | 98.28% | 97.99% |
| Additional allowed | 1.72% | 2.01% |
| PDB ID | 7F71 | 7F8P |

Values in parenthesis are for the highest resolution shell.

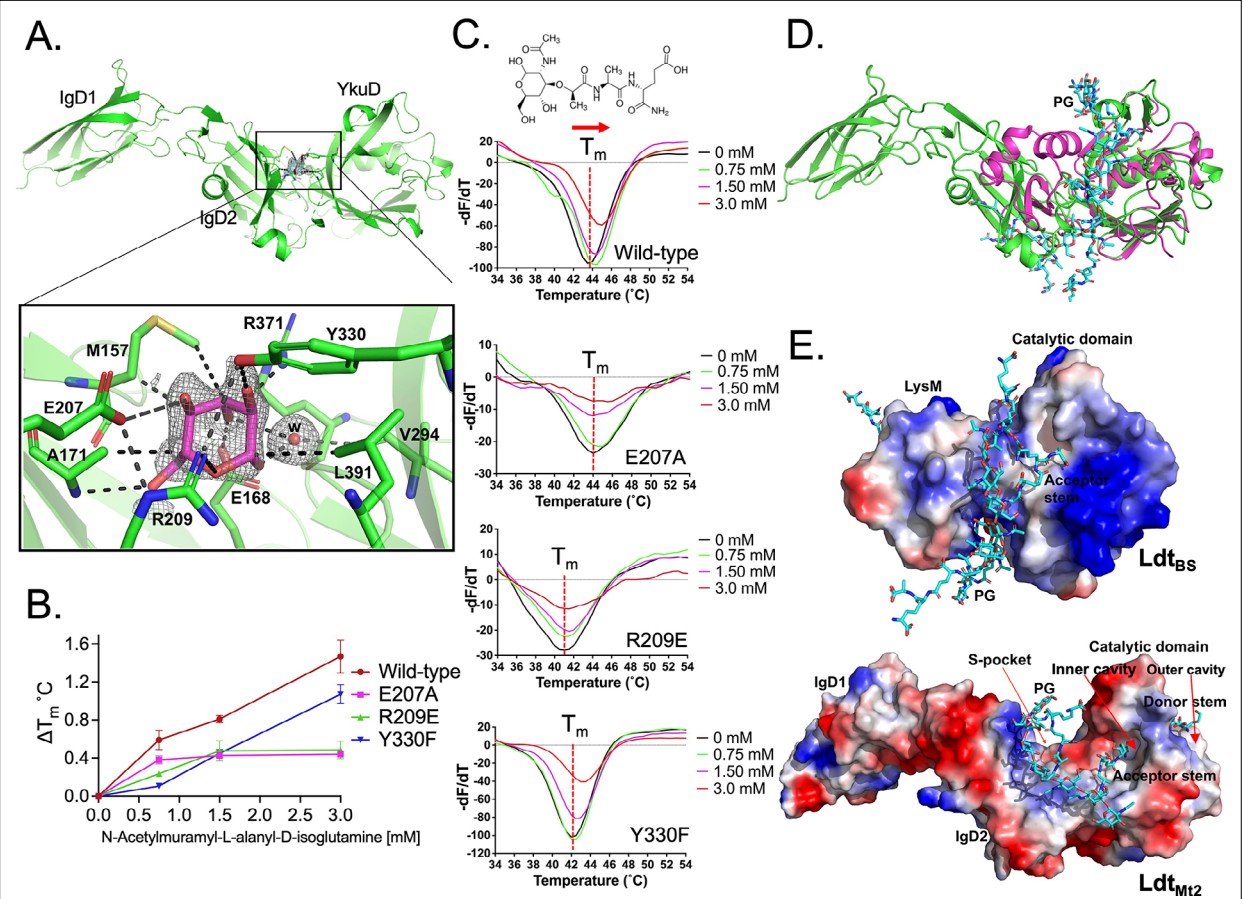

**Figure 1.** Binding studies of peptidoglycan (PG) with Ldt_{Mt2}. (**A**) Crystal structure of Ldt_{Mt2} in complex with one glucose molecule. The inset shows the 2Fo-Fc omit map (contoured at 1.0$\sigma$) of glucose (cyan colour) modelled into the S-pocket of Ldt_{Mt2} in the crystal structure. (**B**) ThermoFluor assay for binding studies with the PG-precursor *N*-acetylmuramyl-L-alanyl-D-isoglutamine hydrate with wild-type Ldt_{Mt2}, R209E, E207A, and Y330F mutants. A change in melting temperature ($\Delta T_m$) at *y*-axis was plotted against the ligand concentrations at *x*-axis in GraphPad Prism software. (**C**) Differential fluorescence ($-dF/dT$) graphs of ThermoFluor assay for Ldt_{Mt2} and mutants. The dotted line indicates the $T_m$, and a red arrow indicates the direction of thermal shift. A chemical structure above the ThermoFluor assay graph is *N*-acetylmuramyl-L-alanyl-D-isoglutamine hydrate. (**D**) Superposition of Ldt_{Mt2} (green) with PG-bound Ldt_{Bs}, the *Bacillus subtilis* L,D-transpeptidase (PDB ID: 2MTZ) (pink). YkuD domain of Ldt_{Mt2} was superposed with catalytic domain of ldt_{BS} with an RMSD of 1.46 Å. PG chain is shown in cyan colour. (**E**) Modelling of PG (cyan colour) into the Ldt_{Mt2} (green). Electrostatic potential (negative in red, positive in blue) highlights the acidic and positively charge surface and binding of PG chain in L,D-transpeptidases Ldt_{BS} and Ldt_{Mt2}.

The online version of this article includes the following source data and figure supplement(s) for figure 1:

**Source data 1.** Raw data on ThermoFluor study of peptidoglycan (PG) substrate binding with wild-type and mutants.

**Figure supplement 1.** 2Fo-Fc map of sugar bound into the S-pocket of Ldt_{Mt2} in the crystal structure.

provides insights into inhibition by β-lactams and supports the context for structure-based design of anti-tubercular drugs.

## Results

### A pocket remote from the catalytic site of Ldt_{Mt2} binds PG

The crystal structure of Ldt_{Mt2} was solved at 1.57 Å resolution (*Table 1*). This high-resolution structure is an improvement of our previous efforts (*Erdemli et al., 2012*; *Kumar et al., 2017*) that enabled us to identify an electron density in a pocket between the IgD2-YkuD domains, and a glucose molecule could be modelled into the electron density at 1.0 sigma (*Figure 1A*, *Figure 1—figure supplement 1*). This glucose molecule is likely to be part of a PG disaccharide moiety originating from the *E. coli* cell lysate during Ldt_{Mt2} purification. The sugar molecule is ensconced at the IgD2-YkuD domain interface in a pocket, which we referred to as the S-pocket, making several electrostatic interactions

with residues R209, E207, E168, R371, Y330, and A171. Three residues M157, A171, and L391 stabilize the sugar through hydrophobic interactions. To provide additional evidence for the binding of PG substrates within the S-pocket, we performed ThermoFluor assays with *N*-acetylmuramyl-L-alanyl-D-isoglutamine hydrate, a precursor of PG. A higher molar concentration of *N*-acetylmuramyl-D-isoglutamine gradually shifted the melting curve of $Ldt_{Mt2}$ indicative of saturable binding behaviour (*Figure 1B, C*, *Figure 1—source data 1*). R209E and E207A mutations in the S-pocket disrupted the binding of *N*-acetylmuramyl-L-alanyl-D-isoglutamine with $Ldt_{Mt2}$. Y330F mutation also affected the binding of PG precursor. This was the order of magnitude of detrimental effect of mutations in the S-pocket on PG precursor binding: R209E ≥ E207A > Y330. As the binding of PG precursor was in mM range, this indicated a weak binding with the S-pocket.

An atomic model of the L,D-transpeptidase from *B. subtilis* ($Ldt_{BS}$) in complex with nascent PG chain was described earlier (PDB ID: 2MTZ) (*Schanda et al., 2014*). As $Ldt_{BS}$ has a LysM domain that $Ldt_{Mt2}$ lacks, we superposed the catalytic domain of $Ldt_{BS}$ with YkuD domain of $Ldt_{Mt2}$ with an RMSD of 1.46 Å. Superposition of the structures of the $Ldt_{Mt2}$–sugar complex with the $Ldt_{BS}$–PG complex suggests that longer nascent PG chains thread across the S-pocket in between the IgD1-YkuD domains of $Ldt_{Mt2}$ (*Figure 1D*). In the $Ldt_{BS}$ structure, the acidic sugar moieties of PG chain bind in-between the LsyM domain and catalytic domain across a positively charged groove (*Figure 1E*). Based on the structural details of PG binding in $Ldt_{BS}$ (*Schanda et al., 2014*) and $Ldt_{Mt2}$ (*Figure 1A*), a PG chain was computationally placed over a positively charged surface across the IgD2-YkuD domain interface

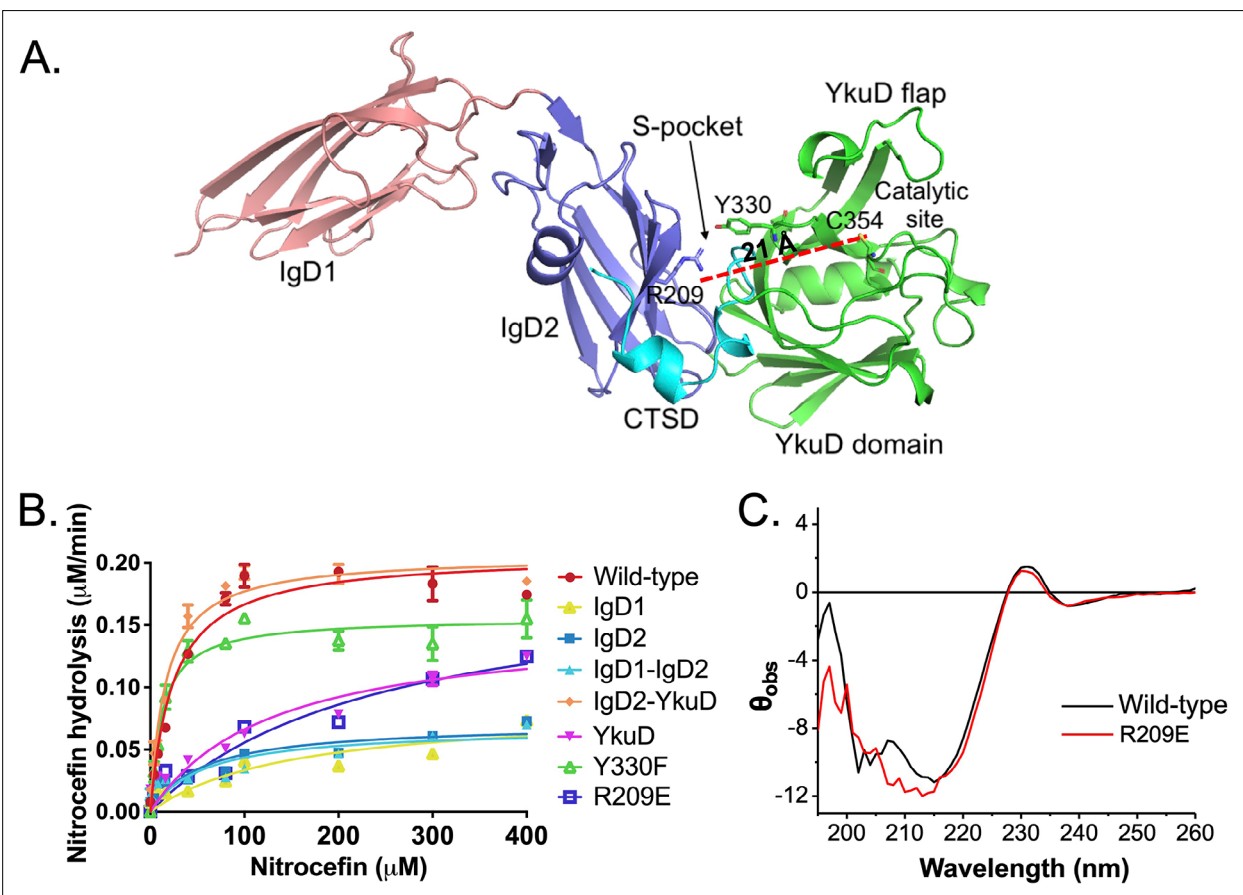

**Figure 2.** Role of the S-pocket in β-lactam hydrolysis. (**A**) The structure of $Ldt_{Mt2}$ with each domain highlighted: IgD1 (orange), IgD2 (blue), YkuD domain (green), and C-terminal subdomain (CTSD) (cyan). A red dotted line demarcates the 21 Å distance between the S-pocket and the catalytic site. (**B**) Chromogenic nitrocefin hydrolysis activity of truncated $Ldt_{Mt2}$ fragments corresponding to the IgD1, IgD2, IgD1–IgD2, YkuD, IgD2-YkuD domains, R209E, and Y330F mutants. (**C**) Circular dichroism (CD) spectra of wild-type and R209E mutant.

The online version of this article includes the following source data and figure supplement(s) for figure 2:

**Source data 1.** Raw data on nitrocefin hydrolysis assay.

**Figure supplement 1.** Chromogenic nitrocefin hydrolysis activity of wild-type $Ldt_{Mt2}$ and E207A mutant.

**Table 2.** Kinetic parameters of β-lactam hydrolysis by $Ldt_{Mt2}$ and mutant proteins.

| Enzyme | $V_{max}$ (µM/min) | $K_m$ (µM) | $K_{cat}$ (s$^{-1}$) | $K_{cat}/K_m$ (M$^{-1}$ s$^{-1}$) |
|---|---|---|---|---|
| $Ldt_{Mt2}$ (ΔN55) | 0.23 ± 0.01 | 16.32 ± 1.78 | 7.7E−4 | 47.18 |
| IgD1 | Ambiguous | – | – | – |
| IgD2 | Ambiguous | – | – | – |
| IgD1–IgD2 | Ambiguous | – | – | – |
| IgD2-YkuD | 0.21 ± 0.01 | 16.18 ± 2.24 | 7.0E−4 | 43.26 |
| YkuD | 0.15 ± 0.02 | 129.5 ± 41.80 | 5.0E−4 | 3.86 |
| R209E | 0.25 ± 0.07 | 428.40 ± 195.9 | 8.3E−4 | 1.90 |
| Y330F | 0.15 ± 0.01 | 11.12 ± 3.15 | 5.0E−4 | 44.96 |
| S351A | 0.11 ± 0.02 | 123.1 ± 56.7 | 3.7E−4 | 3.01 |
| C354A | Ambiguous | – | – | – |
| H352A | 0.10 ± 0.01 | 21.14 ± 6.11 | 3.3E−4 | 15.61 |
| H336N | 0.07 ± 0.01 | 39.13 ± 17.92 | 2.3E−4 | 5.88 |
| M303A | 0.14 ± 0.01 | 11.71 ± 2.90 | 4.7E−4 | 40.14 |
| S337A | 0.23 ± 0.01 | 23.30 ± 2.51 | 7.6E−4 | 32.62 |
| K282A | 0.16 ± 0.01 | 12.06 ± 3.23 | 5.3E−4 | 43.94 |

encompassing the S-pocket. This computational modelling of a longer PG chain spatially aligns one of its tetrapeptide stem across an inner cavity of the catalytic site in YkuD domain in $Ldt_{Mt2}$ (*Figure 1E*), similar to reports of carbapenem and a PG-moiety binding in the same position (*Bianchet et al., 2017*; *Erdemli et al., 2012*; *Kumar et al., 2017*). This inner cavity of the catalytic site is proposed to bind the acceptor tetrapeptide stem, and the outer cavity to bind the donor tetrapeptide stem prior to their 3–3 transpeptide cross-linkage by $Ldt_{Mt2}$ (*Erdemli et al., 2012*). Based on our crystal structure and modelling study, we propose that the S-pocket anchors the disaccharide moiety of one of the nascent PG chains prior to transpeptidation of tetrapeptide stems in the catalytic site.

### The S-pocket modulates β-lactam hydrolysis activity

Our crystal structure reveals that $Ldt_{Mt2}$ is composed of three distinct domains as shown in *Figure 2A*. As the S-pocket resides within the IgD2-YkuD domain interface, we investigated whether the S-pocket or different $Ldt_{Mt2}$ domains play contributing role in the enzyme's catalytic function. Due to the lack of tractable enzymatic assays with native PG substrates for observing physiological catalytic activity, we choose nitrocefin, a chromogenic β-lactam, as a reporter substrate to assess the β-lactam hydrolysis activity (*Kumar et al., 2017*). To undertake this study, we expressed and purified fragments of $Ldt_{Mt2}$ corresponding to IgD1, IgD2, IgD1–IgD2, IgD2-YkuD, and YkuD domains. The full-length $Ldt_{Mt2}$ holo-enzyme showed a $V_{max}$ of 0.23 µM/min and $K_m$ of 16.32 µM in the nitrocefin hydrolysis assay (*Table 2*). Deletion of the IgD1 domain assessed by the IgD2-YkuD domain fragment resulted in no effect on the β-lactam hydrolysis. However, deletion of IgD2 from the YkuD domain as assessed by the YkuD fragment alone led to a significant adverse effect on the nitrocefin hydrolysis activity with an increase in the Km value to 129 µM (an ~eightfold increase) and a decline in enzyme turn-over by ~tenfold (*Figure 2B*, *Figure 2—source data 1*, and *Table 2*). This suggests an important role of S-pocket which is partly carried by the IgD2 domain in governing the catalytic activity of $Ldt_{Mt2}$ enzyme. We further evaluated the role of the S-pocket by generating site-directed mutations at residues R209, Y330, and E207 as they are situated within the S-pocket and interact with the sugar moiety (*Figure 2B*). R209 makes part of IgD2 domain and Y330 residue comes from the YkuD domain into the S-pocket. Y330F mutation led to a decrease in the nitrocefin hydrolysis with a $V_{max}$ 0.15 µM/min and $K_m$ of 11.12 µM. R209E mutant hydrolysed nitrocefin with a $K_m$ 428 µM that is ~26-fold higher than wild-type, while its $V_{max}$ remained almost the same as wild-type. A high $K_m$ value is an indicator of weak binding of the substrate with the enzyme, and shows that the enzyme would need a greater number of substrate

molecules to achieve a maximum rate of reaction. A E207A mutation also showed a decrease in the β-lactam hydrolysis activity as compared to wild-type (*Figure 2—figure supplement 1*). It is highly likely that the S-pocket has a role in modulating the catalytic activity of Ldt$_{Mt2}$ enzyme allosterically. We ruled out any impact of R209E mutation on the secondary structure content of Ldt$_{Mt2}$ enzyme using circular dichroism (CD) spectroscopy (*Figure 2C*). The recorded difference in CD spectra is not significant to suggest any structural changes or problem in protein folding due to R209E mutation. In fact, an overall good overlapping CD spectrum of the wild-type and the R209E mutant suggests that the proteins maintain an overall similar folds. Moreover, the changes recorded in the CD spectra are gross overall changes in the structural element, so even if there are some minor changes in the CD spectra at 210 nm, it is difficult to correlate the region of change. Our biophysical ThermoFluor assay also suggests no major difference in the thermal melting of R209E and wild-type, indicating no major changes in overall fold of the protein (*Figure 1C*).

## The S-pocket cross-talks with the catalytic site to modulate β-lactam hydrolysis

To evaluate the effects of mutations in the S-pocket on catalytic site activity ~21 Å away, we ran molecular dynamic (MD) simulations of the Ldt$_{Mt2}$ wild-type and R209E mutant proteins. Changes in the structural features, before and after the R209E mutation were analyzed by calculating α-alpha RMSD (root mean square deviation) over the course of 200-ns simulation. These calculations were performed using *GROMACS* (*Pronk et al., 2013*). While the mutant model exhibits slightly higher dynamics after 65 ns, the overall RMSD of the models ranged from 0 to 0.45 Å and showed stable conformation for the entire period. Both models show constant stability after 120 ns (*Figure 3—figure supplement 1*). After 200 ns of MD simulations, structural and conformational changes were observed in the catalytic center of YkuD domain including the catalytic triad residues C354, H336, and S337 and other residues in the catalytic site, namely, S351, M303, and W340. After the MD simulations in wild-type Ldt$_{Mt2}$, H336-NE2 formed a hydrogen bond interaction with side chain hydroxyl group oxygen (OG) of S351 (*Figure 3A*). A probably density graph calculated a close hydrogen bond distance between H336-NE2 and S351-OG (*Figure 3B*, *Figure 3—source data 1*). A hydrogen bond interaction was also observed between H336-ND1 and backbone oxygen (O) of S337 (*Figure 3A*), and this interaction has been reported to be important for stabilizing the tautomer of H336 protonated at NE2 (*Erdemli et al., 2012*).

When we assessed the MD simulated structure of the R209E mutant, we found that imidazole ring of H336 was flipped that led to a disruption of hydrogen bond interaction between H336-ND1 and backbone oxygen (O) of S337 (*Figure 3A*). A probably density graph also revealed no hydrogen bond interaction between H336-ND1 and S337-O in R209E mutant (*Figure 3B*, *Figure 3—source data 1*). In addition, no hydrogen bond interaction was found between H336-NE2 and the hydroxyl group oxygen (OG) of the S351 residue and a probably density graph calculated between H336-NE2 and S351-OG pair also computed an increase in their distance during 200 ns of simulations (*Figure 3B*, *Figure 3—source data 1*). Another W340 residue that resides outside the catalytic pocket comes closer to M303 and H336 through hydrophobic interactions to block the outer pocket of catalytic core. A close distance between W340-Çα and M303-Çα was also confirmed by probably density graph calculated along 200 ns of MD simulations (*Figure 3B*, *Figure 3—source data 1*). Such blockage of outer catalytic pocket by W340 would also hinder the dynamics of the YkuD flap, which has been reported to be important in β-lactam binding (*Fakhar et al., 2017*).

We further computed the root mean square fluctuation (RMSF) of wild-type and R209E mutant at 200 ns MD simulations to predict areas of major fluctuations in the structure of Ldt$_{Mt2}$ and an impact of R209E mutation (*Figure 3—figure supplement 2A*). Notably, R209E mutation led to these major changes in the fluctuations in Ldt$_{Mt2}$: (1) an increase in the fluctuations of a loop that interacts with S351 residue, (2) a decrease in the fluctuations of YkuD flap, and (3) a decrease in the fluctuations in the catalytic center. We further delved into the structural changes that were propagating between the S-pocket and catalytic center through the fluctuating regions. We selected a number of residue pairs in the fluctuating regions that were most impacted by R209E mutation (*Figure 3—figure supplement 2B*). A dynamic distance and probability density graph of Y330-OH/D393-OD1 (a residue pair in S-pocket) computed a decrease in their dynamic distance subsequent upon R209E mutation, and this was an indication of a tighter junction between the YkuD domain and S-pocket. A Y292-OH/A288-N

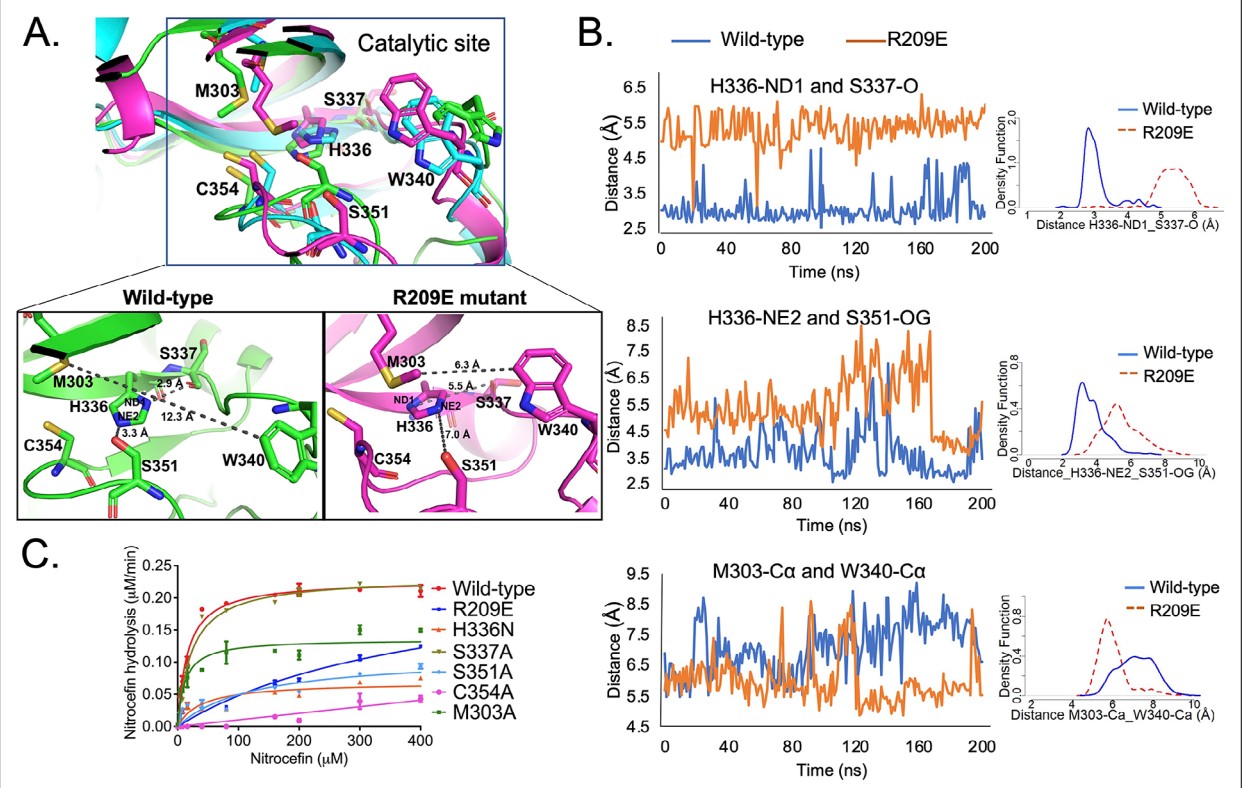

**Figure 3.** S-pocket crosstalk with the catalytic site of Ldt$_{Mt2}$. (**A**) Superposition of molecular dynamic (MD) simulated structures of catalytic site of wild-type Ldt$_{Mt2}$ (146–408 residues, green) and the R209E mutant (146–408 residues, pink) at 150 ns trajectory along with a trajectory (cyan) at 0 ns. The inset shows a detailed view of the catalytic site of the wild-type protein and R209E mutant at 150 ns trajectory. (**B**) Dynamic distance analysis of key residue pairs vs. simulation time calculated from 200 ns of MD simulation run. A density function graph is also plotted. (**C**) Chromogenic nitrocefin hydrolysis activity of wild-type Ldt$_{Mt2}$ and different mutants with alterations in both the S-pocket and catalytic site.

The online version of this article includes the following source data and figure supplement(s) for figure 3:

**Source data 1.** Raw data on dynamic distance network analysis.

**Source data 2.** Raw data on nitrocefin hydrolysis assay.

**Source data 3.** Raw data on network analysis of wild-type Ldt$_{Mt2}$ and R209E mutant.

**Figure supplement 1.** RMSD graph of wild-type Ldt$_{Mt2}$ (blue) and R209 mutant (orange) over the duration of 200 ns of molecular dynamic (MD) simulations.

**Figure supplement 2.** An analysis of dynamics in-between the S-pocket and catalytic site in Ldt$_{Mt2}$.

**Figure supplement 3.** Network analysis of Ldt$_{Mt2}$ dynamics.

pair at the interface with S-pocket in the YkuD domain also became more closer with a hydrogen bond distance to tightly hold a loop that extended towards the catalytic site. A residue K282 in the loop made hydrogen bond interaction with the backbone oxygen (O) of catalytic site residue S351, however, R209E mutation led to an increase in their dynamic distance. This increase in the dynamic distance of S351-O/K282-N pair could impact catalysis, as S351 was found important for stabilizing H336-NE2 in the catalytic center (*Figure 3A, B*). And indeed, a K282A mutation led to a decrease in the enzymatic activity with a $V_{max}$ 0.16 µM/min and $K_m$ ~ 12 µM (*Table 2*). R209E mutation also led to the structural changes into the YkuD flap; H300 made more close interactions with D321-OD1 and D323-OD1 to render the flap less dynamic and fluctuating, as evident in the RMSF graph. Another pair of residues, D304-OD1/S306-OG, became more closer with a hydrogen bond distance to impact the YkuD flap fluctuations and dynamics in R209E mutant. A decrease in the fluctuations of YkuD flap also impacted the dynamic distance of Y308 residue with Q327 in R209E mutant, as Y308 resided on the tip of YkuD flap. In wild-type, dynamic distance of Y308-Q327 pair was found multiphasic that was an indication of movements of YkuD flap over the catalytic site, however, this dynamic distance became monophasic subsequent upon R209E mutation. These findings suggest that R209E mutation impacts

the fluctuations and dynamics of YkuD flap over the catalytic site. Based on the dynamic distance analysis, we propose two major allosteric modulations in Ldt$_{Mt2}$ that can be regulated by S-pocket: (1) modulation of S351 residue via K282 to stabilize H336-NE2 in the catalytic site; (2) modulation of YkuD flap dynamics via Y330, Q327, D304, S306, and H330 to impact the catalysis. We suggest that R209E mutation in the S-pocket impacts both of these allosteric modulations to slow down the catalysis, as seen in nitrocefin hydrolysis assay (*Figure 2B*).

To identify the functional relevance of dynamic distance that were computed in wild-type and R209E mutant, we performed in vitro β-lactam hydrolysis activity with site-directed mutants of the catalytic triad residues C354, H336, and S337, and other important residues namely M303 and S351 that were highlighted in MD simulations. Mutation of catalytic triad residue S337 to alanine did not disrupt the β-lactam hydrolysis activity and this is expected as a S337A mutation should not disrupt the hydrogen bond interaction between H336-ND1 and backbone oxygen of alanine residue. The S337 residue has been reported as an important part of catalytic triad (C354-H336-S337) in Ldt$_{Mt2}$ (*Erdemli et al., 2012*). In L,D-transpeptidase from *B. subtilis*, cysteine–histidine–glycine makes a catalytic triad where serine (in Ldt$_{Mt2}$) corresponds to glycine (in Ldt$_{BS}$) (*Lecoq et al., 2012*). A hydrogen bond interaction with H336-ND1 should remain conserved even if S337 is replaced by alanine or glycine. Mutation of C354 to alanine and H336 to asparagine disrupted β-lactam hydrolysis as expected. Additionally, mutation of the S351 residue to alanine disrupted β-lactam hydrolysis activity, almost to the same degree as seen in the H336N mutant, with a ~15-fold decrease in enzyme turn-over (*Table 2*). In MD simulation runs with the wild-type structure, the S351 sidechain hydroxyl group forms hydrogen bond interaction with H336-NE2, and this hydrogen bond interaction is absent in the R209E mutant structure (*Figure 3A*). H336 is important for deprotonating the C354 sulphur to allow nucleophilic attack on carbonyl group of β-lactam ring (*Erdemli et al., 2012*). We suggest that, in addition to C354-H336-S337 catalytic triad, S351 residue may also form an important part of catalytic center to stabilize H336 during β-lactam binding and hydrolysis in Ldt$_{Mt2}$.

## S-pocket cross-talks with catalytic site through an allosteric communication pathway

We further delved into the pathways of communication signals that propagate between the S-pocket and catalytic site. Network analysis was performed using correlation map, shortest distance paths and centrality analysis between the amino acids of Ldt$_{Mt2}$ enzyme (*Figure 3—figure supplement 3* and *Figure 3—source data 3*). A degree of coupled motion in the Ldt$_{Mt2}$ was measured by normalizing the cross-correlation matrix of atomic fluctuations over the length of 200-ns simulations. There is a coupling observed between R209 amino acid and distant catalytic site residues showed as white boxes (*Figure 3—figure supplement 3A*). Degree of nodes as centrality were calculated through which the major traffic of communication signals propagates over the length of 200-ns simulations (*Figure 3—figure supplement 3B*). This highlighted a number of important residues (nodes here) in the S-pocket (namely R209, Y330, and Q327), YkuD flap (namely I301, D304, S305, D321, V322, Y308, Y318, and M303) and the catalytic center (namely H337, C354, L355, S351, S337, and W340) with high degree of centrality.

Furthermore, an examination of edge betweenness centrality revealed shortest paths through which the communication signals propagate (*Figure 3—figure supplement 3C*). One shortest communication path with high edge-betweenness (>6 threshold) is R209 > Y330 > I328 > L355 that starts at S-pocket and ends at the catalytic center. Other shortest paths from R209 via Y330 also pass communication signals directly through the YkuD flap to reach the catalytic site. A single arginine 209 to glutamate mutation increases the path length of communication between the S-pocket and catalytic site (*Figure 3—figure supplement 3D*), indicating the key role of R209 residue in conservation of the shortest pathways. Residues that greatly affect the path length upon mutation have been hypothesized to be important for the allosteric communication (*del Sol et al., 2006*).

Based on the network analysis, we suggest two major allosteric communication pathways that emanate from the S-pocket: (1) direct-allosteric communication pathway to impact the dynamic motion of the catalytic center; (2) indirect-allosteric communication via the YkuD flap (*Figure 3—figure supplement 3E*). Direct-allosteric communication emanates through R209 > Y330 > I328 > L355 to impact the dynamic motion of H336 residue. I328 interacts with L355 through the hydrophobic interactions. L355 backbone oxygen makes hydrogen bond interaction with H336-NE2. As H336 forms an

important part of catalytic triad (*Erdemli et al., 2012*), any alteration in the direct-allosteric communication pathway via I328/L355 nodes may impact the catalysis. An indirect-allosteric communication from the S-pocket via YkuD flap involves a number of residues, but role of M303 seems to be most important one as per the network analysis. M303 is the node from where multidirectional communications flow towards catalytic residue H336 and YkuD flap residues Y318, D321, and V322. M303 makes hydrophobic interactions with Y308, Y318, and V322 and these multiple interactions may have a paramount impact on YkuD dynamics and catalytic residue H336 that is stabilized through hydrophobic interaction by V322. We confirmed the importance of M303 residue by site-directed mutagenesis that made a detrimental impact on β-lactam hydrolysis activity by decreasing the $V_{max}$ to 0.14 µM/min and $K_m$ to 11.7 µM (*Figure 3C* and *Table 2*). This is an eye-opening observation that several hydrophobic core nodes relay communications signals to the catalytic center that may couple with catalytically permissive environment. A most remarkable observation in the network analysis is the convergence of most of the communications signals at W340 residue and at H336. A dynamic distance and probability density graph analysis also computed a major dynamic change in W340 residue upon R209E mutation (*Figure 3—figure supplement 2*) and that may impact the catalytic process. A single R209E mutation disrupts the convergence of communication signals to W340 (*Figure 3—figure supplement 3D*) and also its dynamic distance with M303 and H336 residues (*Figure 3A, B*).

## Both the S-pocket and catalytic site participate in β-lactam recognition cooperatively

Among β-lactams, the penicillin and cephalosporin subclasses are readily hydrolysed by Ldt$_{Mt2}$, while the carbapenem subclass inhibits Ldt$_{Mt2}$ by irreversible acylation of C354 residue in the active site. In the current study, we used the carbapenem molecule, biapenem, to evaluate acylation of Ldt$_{Mt2}$ that could be influenced by a relay of of allosteric communications between the S-pocket and catalytic site, as depicted in *Figure 3—figure supplement 3E*. The rate of acylation by biapenem was measured by monitoring a decrease in biapenem absorbance at 292 nm wavelength. A single R209E mutation in the S-pocket completely disrupted biapenem-mediated acylation of the Ldt$_{Mt2}$ enzyme (*Figure 4A, B*, *Figure 4—source data 1*). Mutation of catalytic residues C354, H336, and S351 also abrogated acylation with biapenem. These findings suggest that both the S-pocket and the catalytic center play important roles in driving acylation of the C354 catalytic residue by biapenem.

To further understand the role of the S-pocket and catalytic site in biapenem binding to Ldt$_{Mt2}$, we performed ThermoFluor assays. Different amounts of biapenem (0–400 µM) were titrated into 5.0 µM of Ldt$_{Mt2}$ enzyme, and thermal shifts were measured at different drug concentrations (*Figure 4C, D*, *Figure 4—source data 2*). These studies revealed interesting observations: (1) increasing concentrations of biapenem led to a gradual change in melting temperature of Ldt$_{Mt2}$ until it was fully saturated, and (2) biapenem binding decreased the melting temperature of protein. In the first observation, we found that Ldt$_{Mt2}$–biapenem binding could be saturated only by enzyme:drug ratios as high as 1:80. This strongly suggests that biapenem saturates a surface of Ldt$_{Mt2}$ through reversible, non-covalent interactions, as the covalent interactions have to be with a 1:1 molar ratio with rapid turn-over (in fractions of a second, see *Figure 4B*) and non-reversible. Beyond to its well-known covalent binding at the catalytic site (*Kumar et al., 2017*), these findings are consistent with non-covalent, saturable binding of biapenem to a second surface on Ldt$_{Mt2}$. From the second observation, we conclude that biapenem binding destabilizes the protein possibly through structural changes. This structural destabilization may supersede the well-known structural changes in the YkuD flap at the catalytic site that are known to occur during β-lactam binding and covalent reaction with the SƳ atom of C354 (*Bianchet et al., 2017*; *Fakhar et al., 2017*; *Kim et al., 2013*).

As the S-pocket mutant R209E exhibited diminished β-lactam hydrolysis (*Figure 3C*) and acylation by biapenem (*Figure 4A, B*), we further analyzed the consequence of mutations in the S-pocket on the physical binding of biapenem. In contrast to a gradual decrease in the thermal stability displayed by wild-type Ldt$_{Mt2}$ upon biapenem binding, the R209E mutant showed only a subtle change in Tm, and the saturating property of biapenem was virtually absent even at the highest concentration of 400 µM (*Figure 4C, D*, *Figure 4—source data 2*). E207A and Y330F mutations also showed detrimental effect on saturable behaviour of biapenem. This was the order of magnitude of detrimental effect of mutations in the S-pocket on reversible biapenem binding: R209E > E207A > Y330. We conclude that the mutations in the S-pocket hindered both non-covalent (as seen in

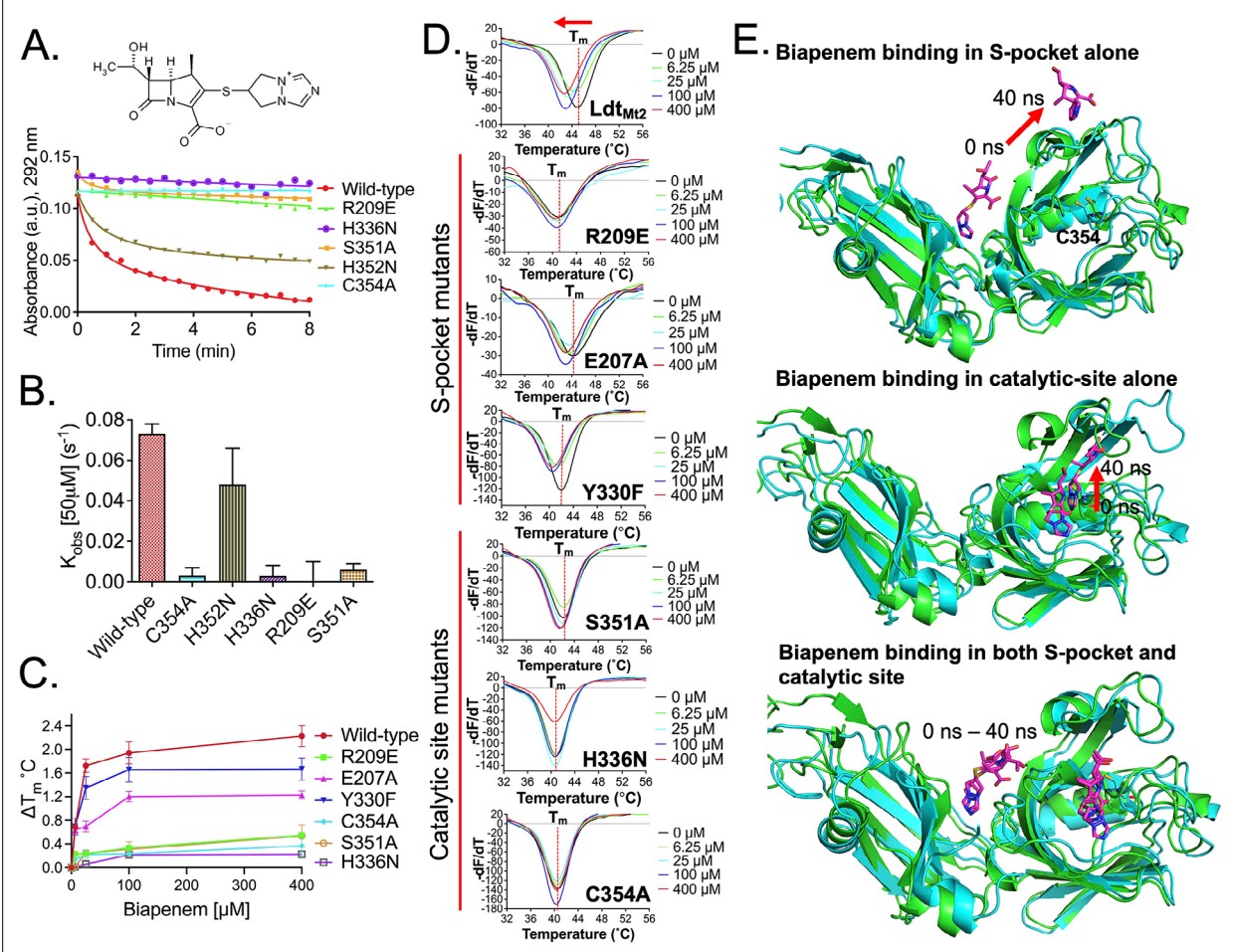

**Figure 4.** Role of the S-pocket and catalytic site in recognizing biapenem. (**A**) Acylation activity of biapenem with Ldt$_{Mt2}$ and mutants R209E, H336N, S351A, H352N, and C354A was monitored at 292 nm wavelength using UV–visible spectrophotometry. Maximum absorbance spectra of biapenem were found at 292 nm that was used to monitor decrease in biapenem concentration upon acylation with the Ldt$_{Mt2}$. The chemical structure of biapenem is shown above the biapenem acylation graph. (**B**) Rate of acylation of 50 µM biapenem per second with Ldt$_{Mt2}$ and mutants. (**C**) ThermoFluor assays for binding of biapenem with Ldt$_{Mt2}$ and mutants R209E, E207A, Y330F, S351A, H336N, and C354A mutants. A change in melting temperature ($\Delta T_m$) was plotted at $y$-axis verses the ligand concentrations at $x$-axis in GraphPad Prism software. (**D**) Differential fluorescence ($-dF/dT$) graphs of ThetrmoFluor assay for Ldt$_{Mt2}$ and mutants. The dotted line indicates the $T_m$, and a red arrow indicates the direction of thermal shift. (**E**) Molecular dynamic (MD) simulations of Ldt$_{Mt2}$ in complex with biapenem. Ldt$_{Mt2}$ is represented in cartoon with green colour at 0 ns and cyan colour after running the MD simulations at 40 ns, and biapenem is represented in stick model with pink colour. The red arrow indicates the movement of biapenem to a second position revealed by the MD simulations after 40 ns trajectory.

The online version of this article includes the following source data and figure supplement(s) for figure 4:

**Source data 1.** Raw data for biapenem acylation assay.

**Source data 2.** Raw data from ThermoFluor assay for biapenem binding with wild-type and mutants.

**Figure supplement 1.** Molecular dynamic (MD) trajectory of biapenem bound in S-pocket alone in Ldt$_{Mt2}$–Bia$^S$ structure.

**Figure supplement 2.** Molecular dynamic trajectory of biapenem bound in catalytic site alone in Ldt$_{Mt2}$–Bia$^C$ structure.

**Figure supplement 3.** Molecular dynamic trajectory of dual biapenem bound in both S-pocket and catalytic site in Ldt$_{Mt2}$–Bia$^{S-C}$ structure.

*Figure 4C*) as well as covalent interactions with biapenem (as seen in *Figure 4A, B*). Additionally, as biapenem binds negligibly to the R209E mutant in contrast to wild-type Ldt$_{Mt2}$, we did not observe decreases in the melting temperature of the R209E mutant with added biapenem as would be anticipated via structural changes in the YkuD flap (*Fakhar et al., 2017*). This is further illustrated by our MD simulation results wherein the R209E mutation brings W340 residue closer to the YkuD flap residue M303 and active-site core residue H336 to block access to the outer pocket of the catalytic

site (*Figure 3A*, *Figure 3—figure supplement 2*). These R209E mutation-driven structural changes in the catalytic site and YkuD flap likely account for the inability of biapenem to bind to the R209E mutant of $Ldt_{Mt2}$.

We further tested the hypothesis that the catalytic site also demonstrates an interplay with the S-pocket to indirectly influence non-covalent binding of biapenem. Binding studies of biapenem were performed with catalytic mutants C354A, H336N, and S351A using ThermoFluor assays (*Figure 4C, D*, *Figure 4—source data 2*). Varying concentrations of biapenem (0–400 µM) with the catalytic site mutants did not induce any further significant thermal shift, indicative of negligible or insignificant physical binding of biapenem. Acylation with biapenem at the catalytic site was also diminished upon C354A, H336N, and S351A mutation in $Ldt_{Mt2}$ (*Figure 4A*). These experimental observations indicate that mutations in catalytic site disrupt both non-covalent as well as covalent binding of biapenem to $Ldt_{Mt2}$.

From the experimental results with the wild-type, S-pocket mutants and catalytic site mutants (*Figure 4*), we conclude that biapenem has two modes of binding: (1) covalent binding and (2) saturable, non-covalent binding. As covalent binding is well known at the catalytic site, saturable reversible binding seems to be at the S-pocket. In both of the biapenem-binding modes, our data suggest a cooperative role between S-pocket and the catalytic site, as mutations in any of the sites disrupt the binding (*Figure 4A, C*). Towards identifying the structural basis of cooperativity, we performed docking and MD simulation studies of biapenem with $Ldt_{Mt2}$ (*Figure 4E*). A $Ldt_{Mt2}$ crystal structure (PDB ID: 5DU7) that has the catalytic site within the YkuD flap in closed conformation was chosen for docking with biapenem as this structure will not allow the drug binding in the catalytic pocket. A grid was assigned for docking within a 60 Å radius of the catalytic site residue C354. This blind docking around the catalytic site was performed to rule out any biasness in docking results influenced by the experimental data. We found that biapenem docked well within the S-pocket through its pyrazolo[1,2a][1,2,4]triazolium R3 group with a docking score of −6.3 kcal/mol.

Next, MD simulation experiments were further performed with $Ldt_{Mt2}$ structures having biapenem docked (1) alone in S-pocket, (2) alone in catalytic pocket, and (3) both in S-pocket and catalytic site. In the MD simulations with biapenem docked in S-pocket alone, the drug remained in the pocket for 9 ns of MD trajectory before exiting the pocket (*Figure 4E*). Snapshots of different trajectories of biapenem in the S-pocket are shown in *Figure 4—figure supplement 1*. In MD simulations with biapenem docked in the catalytic pocket alone, the β-lactam core ring fluctuated more during 5–40 ns (*Figure 4E*, *Figure 4—figure supplement 2*).

However, when biapenem molecules were docked in both the S-pocket and catalytic site simultaneously, the pyrazolo[1,2a][1,2,4]triazolium R3 group of biapenem remained ensconced in S-pocket for 0–6 ns, made hydrophobic interactions with Y330 and L391 at 7–15 ns, and its pyrrolidine ring made additional π–π interactions with Y330 at 18–28 ns while remaining in the S-pocket, before finally moving out towards the YkuD flap of the catalytic site after 40 ns (*Figure 4E*, *Figure 4—figure supplement 3A*). In the catalytic site over the simulation interval, biapenem movement fluctuated less this time during 0–40 ns (see snapshots of biapenem trajectory in *Figure 4—figure supplement 3B*). Thus, MD simulations suggest that biapenem binding across the S-pocket surface imposes stability in fluctuations of β-lactam movement in the catalytic site. These MD simulations together with our experimental data support a model in which two biapenem molecules are recognized cooperatively by both the S-pocket and the catalytic site, with non-covalent saturable binding and covalent binding to acylate C354, respectively.

**Table 3.** Docking score of $Ldt_{Mt2}$ with β-lactam compounds in kcal mol$^{-1}$ calculated by Autodock.

| Drug | Binding energy (kcal/mol) |
| --- | --- |
| Ampicillin | −7.1 |
| Oxacillin | −8.3 |
| Cefotaxime | −7.8 |
| Biapenem | −6.3 |
| T203 | −6.9 |

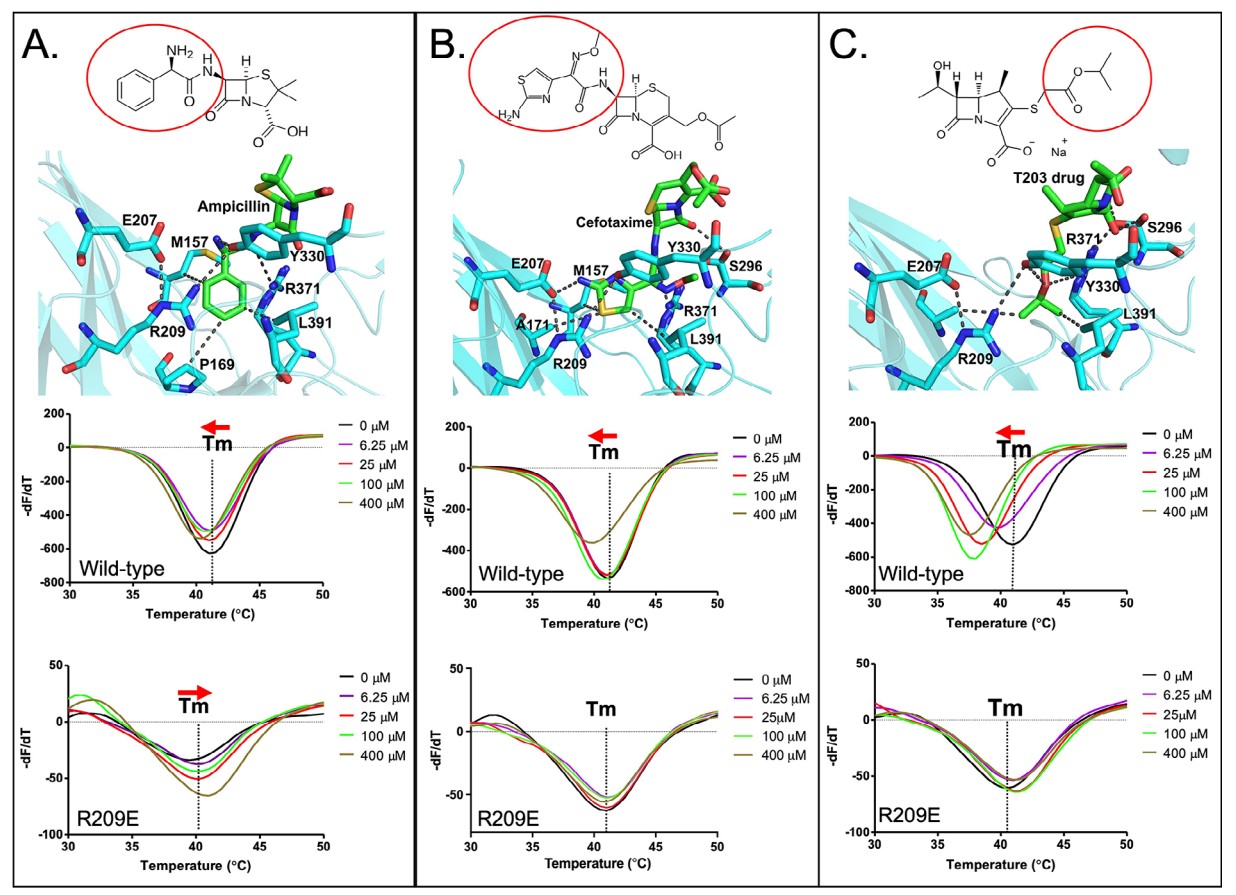

**Figure 5.** Binding of various subclasses of β-lactams to the S-pocket. (**A**) Top: ampicillin (stick model in green) bound to the S-pocket (cyan) of Ldt$_{Mt2}$ through its R1 group side chain, 2-amino-2-phenylacetyl (red oval). Bottom: ThermoFluor assays for binding studies of ampicillin with wild-type Ldt$_{Mt2}$ and the R209E mutant. (**B**) Top: cefotaxime (stick model in green) bound to the S-pocket (cyan) of Ldt$_{Mt2}$ through its R1 group side chain, thiozol-4yl (red oval). Bottom: ThermoFluor assays for binding studies of cefotaxime with wild-type Ldt$_{Mt2}$ and the R209E mutant. (**C**) Top: the experimental carbapenem drug T203 (stick model in green) bound to the S-pocket (cyan) of Ldt$_{Mt2}$ through its R3 group side chain, 2-isopropoxy-2-oxoethyl (red circle). Bottom: ThermoFluor assays for binding studies of T203 drug with wild-type Ldt$_{Mt2}$ and the R209E mutant.

The online version of this article includes the following source data and figure supplement(s) for figure 5:

**Source data 1.** Raw data from ThermoFluor assay for β-lactam binding with wild-type and mutants.

**Figure supplement 1.** Binding of oxacillin drug to the S-pocket.

## Binding patterns of various subclasses of β-lactams in the S-pocket

As Ldt$_{Mt2}$ binds various β-lactams with variable affinities (*Bianchet et al., 2017*), we performed docking studies of various subclasses of β-lactams with the S-pocket. Ampicillin and oxacillin from the penicillin subclass, cefotaxime from the cephalosporin subclass, biapenem from carbapenem subclass and an experimental carbapenem drug, T203, developed by our group (*Kumar et al., 2017*), were chosen for the study. The different β-lactams showed binding with the S-pocket of Ldt$_{Mt2}$ with variable energy scores using Autodock (*Table 3*). Ampicillin docked into the S-pocket with a docking score of −7.1 kcal/mol with its R1 group tail 2-amino-2-phenylacetyl ensconced in the S-pocket through several electrostatic and hydrophobic interactions with the M157, E207, R209, R371, and Y330 residues (*Figure 5A*). Another penicillin subclass member, oxacillin (a penicillinase-resistant penicillin), displayed the highest docking score of −8.3 kcal/mol through its R1 group 5-methyl-3-phenyl-1,2-o xazole-4-carbonyl binding in the S-pocket (*Figure 5—figure supplement 1A*). Cefotaxime docked to the S-pocket with a docking score of −7.8 kcal/mol through R1 group tail thiozol-4yl (*Figure 5B*), similar to the biapenem R3 group (*Figure 4E*). The experimental carbapenem T203 docked to the S-pocket with the least −6.9 kcal/mol binding with its R3 group 2-isopropoxy-2-oxoethyl (*Figure 5C*). The β-lactam ring moieties of all of these β-lactams were found to be free of any interactions with

the S-pocket or surrounding residues, similar to biapenem. However, after 18–28 ns of MD simulation trajectory, the pyrrolidine ring of biapenem could make π–π interaction with Y330 (*Figure 4E*), and it is possible that similar late binding interactions may occur similarly with other β-lactams.

ThermoFluor assays were also performed to investigate the binding behaviours of these additional ß-lactam class members with Ldt$_{Mt2}$ (*Figure 5*, *Figure 5—source data 1*). Ampicillin, which has been reported to be readily hydrolysed by Ldt$_{Mt2}$ (*Bianchet et al., 2017*), showed a saturable binding behaviour (*Figure 5A*), but to a significantly lower degree than biapenem (*Figure 4D*). Surprisingly, with ampicillin the R209E mutation in the S-pocket completely reversed the gradual thermal shift in Ldt$_{Mt2}$ towards a higher Tm indicative of an increase in structural stability in the setting of clearly saturable binding (*Figure 5A*). We interpret this to be consistent with reversible acylation of the C354 residue by ampicillin in addition to S-pocket binding. In support of this, a reversible acylation of the L,D-transpeptidase (Ldt$_{fm}$ from *E. coli*) by β-lactams in the catalytic site has been reported recently (*Edoo et al., 2017*; *Zandi and Townsend, 2021*). Oxacillin also showed a saturable binding with Ldt$_{Mt2}$ (*Figure 5—figure supplement 1B*). With cefotaxime, the R209E mutation in the S-pocket strongly diminished saturable binding. And lastly, binding of experimental carbapenem drug T203 displayed a large saturable thermal shift with Ldt$_{Mt2}$ (*Figure 5C*), similar to biapenem (*Figure 4D*). We conclude that many β-lactams (despite being weak or strong inhibitors of Ltd$_{Mt2}$ activity) bind through the S-pocket with a saturable binding behaviour; however, the carbapenem subclass brings maximum thermal destabilization in protein structure due to non-hydrolysable covalent binding in catalytic site. Other classes of β-lactam drugs, specifically the penicillins and cephalosporins, are known to be readily hydrolysed by Ldt$_{Mt2}$ (*Cordillot et al., 2013*; *Kumar et al., 2017*).

## Structural details of allosteric cooperation in dual β-lactam binding

To further understand the high-resolution details of structural changes may that occur in Ldt$_{Mt2}$ upon covalent binding with β-lactam, the crystal structure of Ldt$_{Mt2}$ was solved in complex with the experimental carbapenem drug T203 at a 1.7 Å resolution. Electron densities were observed in both the S-pocket and the outer cavity of catalytic pocket in the Ldt$_{Mt2}$. *Figure 6—figure supplement 1A, B* shows the Fo-Fc omit map (contoured at 3.0$\sigma$) in both S-pocket and catalytic site. Consistent with our docking results of T203 drug with Ldt$_{Mt2}$ (*Figure 5C*), the 2-oxoethyl side-chain of R3 group from T203 could be modelled into the electron density of the S-pocket. A second T203 drug was also modelled into the electron density map of the catalytic pocket. *Figure 6A, B* shows the 2Fo-Fc electron density map (contoured at 1.0$\sigma$) of T203 modelled in the S-pocket and catalytic site of the Ldt$_{Mt2}$ in the crystal structure.

In the S-pocket of Ldt$_{Mt2}$, the 2-oxoethyl sidechain of T203 drug is stabilized through hydrophobic interactions with the A171, M157, P169, and L390 residues (*Figure 6A*, *Figure 6—figure supplement 1C*). R371 makes an electrostatic interaction with the oxygen of the 2-oxoethyl moiety. No electron density was observed for the pyrrolidine ring of T203, while its carboxylic group fitted into an electron density making electrostatic interactions with backbone nitrogen of S296 and the guanidium side chain of R371. The modelling results of T203 into the electron density of the S-pocket were similar to the docking results of T203 drug that also has R3 group ensconced into the S-pocket with pyrrolidine ring remaining free of any interaction with Ldt$_{Mt2}$ (*Figure 5C*, *Figure 6—figure supplement 1E*).

In the catalytic site, the T203 carbapenem interacts with the outer cavity (*Figure 6—figure supplement 1D*) at a covalent distance from the S$\Upsilon$ atom of C354 (*Figure 6B*). The carbonyl oxygen of T203 makes hydrogen bond interactions with the hydroxyl group of Y318. The electron density for the R1-hydroxy ethyl group was not found, similar to three other related experimental carbapenems T206, T208, and T210 (*Kumar et al., 2017*). The methyl group of the pyrrolidine ring makes hydrophobic interaction with the phenyl ring of Y318. The amino N4 of the pyrrolidine ring makes electrostatic interactions with NE2 of H336 and the backbone amide nitrogen of H352. The carboxyl group at C3 of the pyrrolidine ring makes hydrogen bond interactions with the side chains of W340 and N356. W340 also forms hydrophobic interactions with the 2-oxoethyl tail of T203.

We compared the structure of the Ldt$_{Mt2}$–T203 complex with the C354A catalytic mutant structure (PDB ID: 3TX4) to seek alterations in conformation states of the enzyme around its catalytic site, YkuD flap, and S-pocket upon β-lactam binding. We chose the catalytic mutant structure of Ldt$_{Mt2}$ for structural comparison studies only because the wild-type enzyme usually binds ligands and/or substrates from its recombinant bacterial source during the purification steps (*Erdemli et al., 2012*), including in

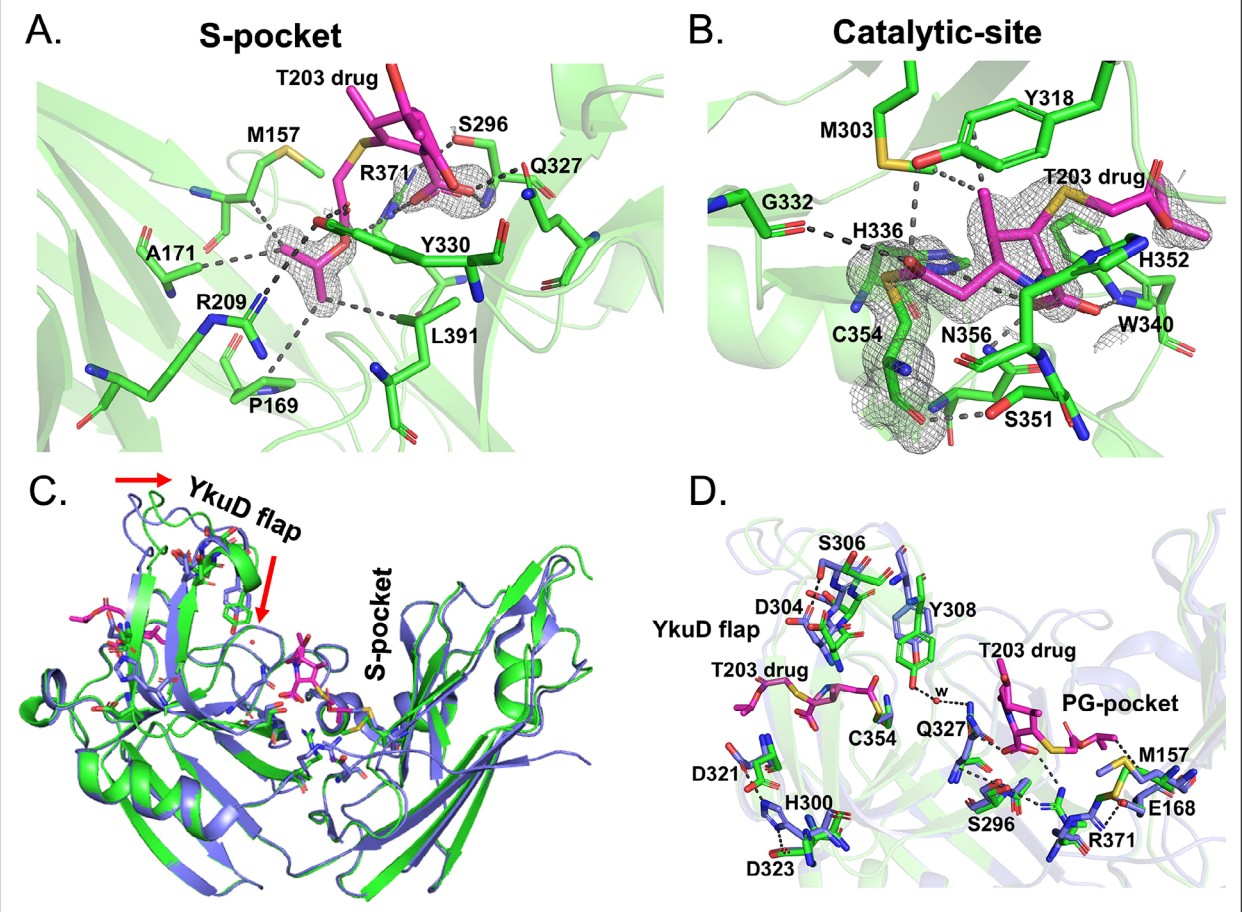

**Figure 6.** Structural studies of Ldt$_{Mt2}$ with the experimental T203 carbapenem drug and allosteric conformation analyses. (**A**) The 2Fo-Fc map (contoured at 1.0$\sigma$) of the T203-R3 group side chain, 2-isopropoxy-2-oxoethyl (pink), modelled in the S-pocket of Ldt$_{Mt2}$ in the crystal structure. (**B**) The 2Fo-Fc omit map (contoured at 1.0$\sigma$) of the full T203 structure (pink) modelled in the catalytic-site of Ldt$_{Mt2}$ where it acylates the C354 residue of Ldt$_{Mt2}$. (**C**) Superposition of the Ldt$_{Mt2}$–T203 complex (green) with C354A catalytic mutant structure (PDB ID: 3TX4, blue). The red arrows indicate movements in YkuD flap upon T203 drug binding. (**D**) Residues that have undergone allosteric alterations upon T203 drug binding are shown with stick models. Ldt$_{Mt2}$–T203 complex residues are represented in green and the C354A catalytic mutant in blue.

The online version of this article includes the following figure supplement(s) for figure 6:

**Figure supplement 1.** Crystal structure studies of binding of T203 drug in the S-pocket and catalytic site in Ldt$_{Mt2}$.

**Figure supplement 2.** RMSD graph of T203 (S-pocket) (red) and T203 (catalytic-pocket) (blue) over the duration of 50 ns of molecular dynamic (MD) simulations in Ldt$_{Mt2}$–T203$^{S-C}$.

the current study. We observed that binding of the T203 drug introduces unique allosteric alterations in the salt bridge and hydrogen bond interactions spanning the entire distance from the S-pocket to the YkuD flap of the catalytic site. Upon T203 drug binding, the YkuD flap bends slightly towards the S-pocket (*Figure 6C*). In the S-pocket, the M157 side chain moves closer to the drug by 1.5 Å to make a hydrophobic interaction with the 2-oxoethyl tail of T203 drug (*Figure 6D*). The R371 residue that was making salt bridge with E168 moves towards S296 through a hydrogen bond interaction and makes an additional ionic interaction with the carboxyl group of the T203 drug. The Q327 side chain that was previously producing a steric conflict with the carboxyl group of T203 drug moves away by a distance of 1.8 Å to make a water-mediated salt bridge with the hydroxyl group of Y308 that also moves down towards the S-pocket by a distance of 2.1 Å. The T203 drug binding induces an additional alteration in the YkuD flap by breaking the hydrogen bond interactions of H300 with D323 as well as D321 and also the interactions between D304 and S306. Breaking of these hydrogen bond interactions possibly relaxes the YkuD flap, enabling it to tilt towards the S-pocket mediated by new water-mediated salt bridge between Y308 and Q327. Alterations in the dynamics of YkuD flap were also observed in MD

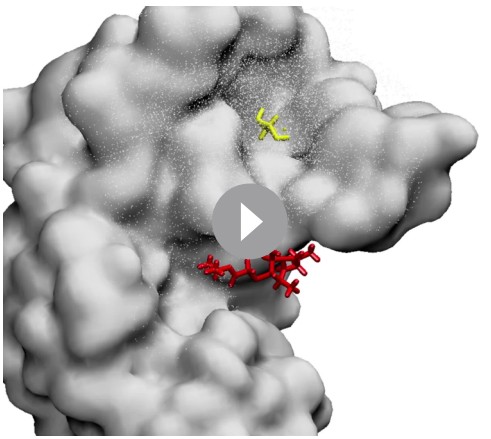

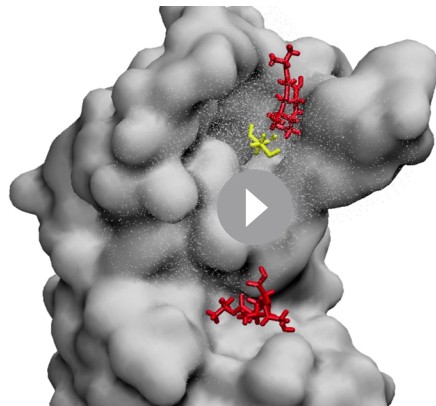

**Video 1.** Visualization of Ldt$_{Mt2}$–T203$^S$ crystal structure in complex with T203 drug at S-pocket alone over the course of 1000 ps equilibration. Residues in and around the catalytic and S-pocket residues are shown as solvent and transparent for proper visualization of the mechanism of the drug. Protein is shown in surface representation as diffused and grey. Cysteine is shown as stick model in yellow.

https://elifesciences.org/articles/73055/figures#video1

**Video 2.** Visualization of Ldt$_{Mt2}$-T203$^{S-C}$ crystal structure in complex with dual T203 molecules bound non-covalently at S-pocket and covalently at catalytic site over 50 ns molecular dynamic (MD) simulation run. Protein and ligands are shown in similar representation as in *Video 1*.

https://elifesciences.org/articles/73055/figures#video2

simulations subsequent upon R209E mutation in the S-pocket (*Figure 3—figure supplement 2*). Several of the allosteric changes observed in the crystal structure were matching with the dynamic distance analysis in MD simulation: (1) distance of H300 with D323 and D321 on YkuD flap, (2) distance of S306 with D304 on YkuD flap, and (3) distance of Y308 with Q327 (*Figure 3—figure supplement 2B*). As these networks of interactions were disrupted by R209E mutation in the S-pocket (*Figure 3—figure supplement 2*) or C354 mutation in the catalytic residue in the crystal structure (PDB ID: 3TX4), this suggests the pathway of allostery communication to be cooperative between S-pocket and catalytic site, and this is quite supported by the experimental data (*Figures 3 and 4*).

In the crystal structure study of Ldt$_{Mt2}$–T203 complex, 2Fo-Fc map shows low occupancy at S-pocket (*Figure 6A*). With low occupancy, the modelling of T203 drug could be biased in the S-pocket. To rule out any biasness and also further validate the cooperativity between the S-pocket and catalytic pocket in dual ß-lactam binding, MD were performed for (1) crystal structure of Ldt$_{Mt2}$ complexed with T203 drugs at both S-pocket and catalytic site (referred to as Ldt$_{Mt2}$–T203$^{S-C}$), and (2) crystal structure of Ldt$_{Mt2}$ complexed with T203 drug at S-pocket alone (referred to as Ldt$_{Mt2}$–T203$^S$, and this structure was prepared by removing T203 drug from the catalytic site in the crystal structure). It was observed that T203 drug in Ldt$_{Mt2}$–T203$^S$ structure left from the S-pocket during the equilibration at 650 ps (*Video 1*). Production run for Ldt$_{Mt2}$–T203$^{S-C}$ was performed for 50 ns (*Figure 6—figure supplement 2*). During 50 ns of simulations, one T203 drug remained bound to S-pocket, while the other one remained covalently attached to C354 residue in catalytic site (*Video 2*). This indicates a cooperativity in-between these the two sites in β-lactam binding and this is in agreement with the MD studies of Ldt$_{Mt2}$ with biapenem drug (*Figure 4E*). Mutational changes in the S-pocket or the catalytic pocket nullify all the allosteric communications that are otherwise important in cooperative binding of dual β-lactams. The consequences of these mutational changes were experimentally confirmed by β-lactam hydrolysis assays (*Figures 2B and 3C*), acylation by biapenem (*Figure 4A, B*) and ThermoFluor assays (*Figures 4 and 5*) with different subclasses of β-lactams.

## Discussion

In addition to the role of L,D-transpeptidases in remodelling the PG in non-replicating *M. tb* (*Lavollay et al., 2008*), this enzyme class is responsible for the resistance of *M.tb* to most β-lactam drugs, except carbapenems (*Cordillot et al., 2013*; *Gupta et al., 2010*). The molecular mechanisms and physiological function of L,D-transpeptidases and the basis for their genetic susceptibility to selective

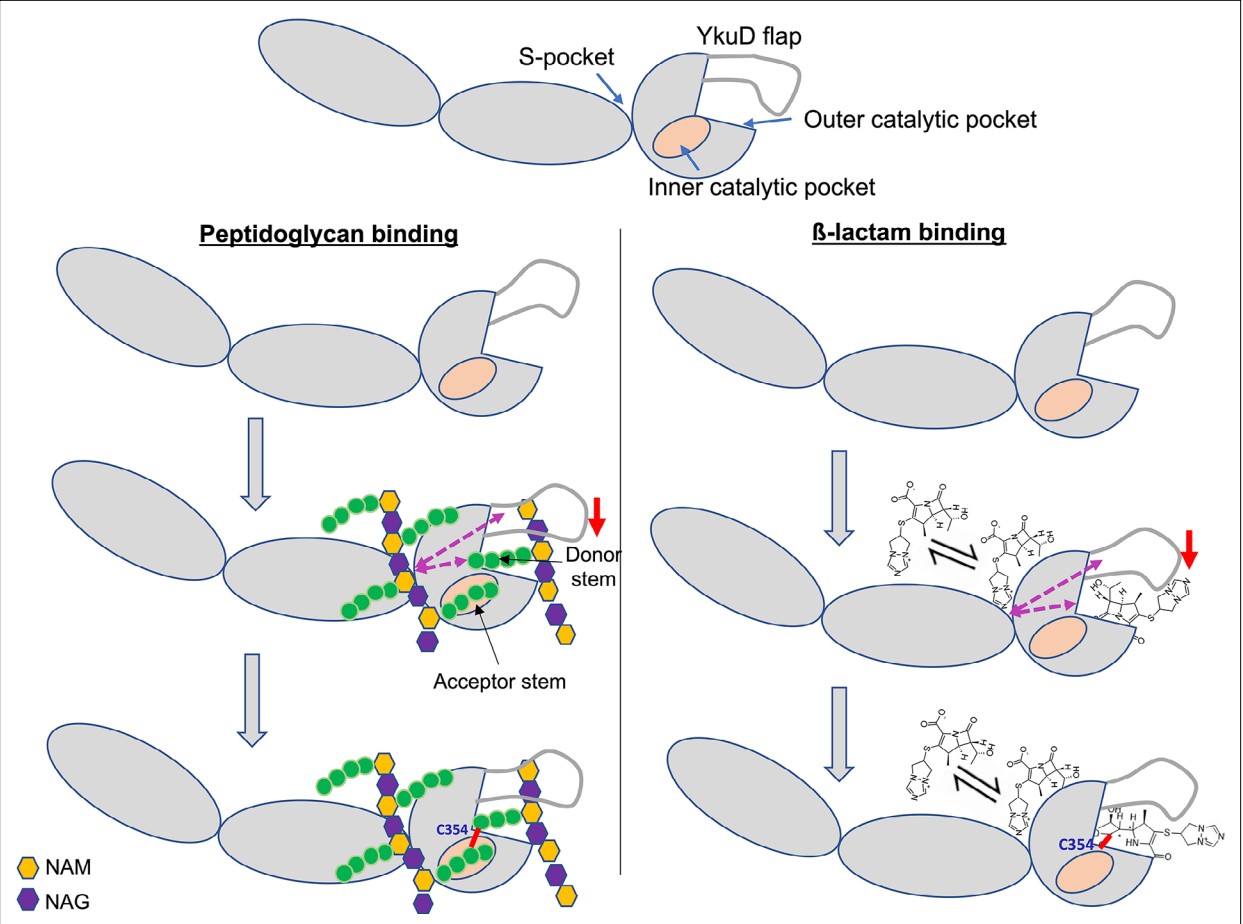

**Figure 7.** Model of dual β-lactam and/or dual peptidoglycan (PG) substrate binding in S-pocket and the catalytic site of Ldt$_{Mt2}$ enzyme that is allosteric cooperative. Purple dotted arrows indicate pathways of allosteric communication and red arrow indicates movement of YkuD flap during β-lactam and/or PG binding. A small red line in the figure indicates a covalent bond between donor and acceptor stem peptides of PG chain or covalent bond between catalytic residue C354 and β-lactam. β-Lactam molecule indicated in the model is biapenem.

The online version of this article includes the following figure supplement(s) for figure 7:

**Figure supplement 1.** S-pocket in different L,D-transpeptidases from Mycobacterium tuberculosis.

β-lactams remains incompletely understood. In this study we reveal important aspects of the physiological function of the *M.tb* L,D-transpeptidase enzyme, Ldt$_{Mt2}$, identify a new PG disaccharide moiety-binding pocket (named the S-pocket), and describe the S-pocket's role in allosteric modulation of the transpeptidase active site. Additionally, we observe that various β-lactams bind to the S-pocket through their tail regions to bring about allosteric changes which predispose the catalytic site for covalent inactivation by a second β-lactam. Based on our findings, we propose a model of dual β-lactam and/or dual PG substrate binding in the S-pocket and the catalytic site of Ldt$_{Mt2}$ and that is allosteric cooperative (*Figure 7*).

*M.tb* contains several paralogs of L,D-transpeptidases, namely Ldt$_{Mt1}$, Ldt$_{Mt2}$, Ldt$_{Mt3}$, Ldt$_{Mt4}$, and Ldt$_{Mt5}$ (*Gupta et al., 2010*). Crystal structures of Ldt$_{Mt1}$ (*Correale et al., 2013*), Ldt$_{Mt2}$ (*Erdemli et al., 2012*), Ldt$_{Mt3}$ (*Libreros-Zúñiga et al., 2019*), and Ldt$_{Mt5}$ (*Brammer Basta et al., 2015*) have been solved and reported to date. All of these paralogs contain a pocket similar to the S-pocket found in Ldt$_{Mt2}$. Corresponding to the R209 residue position in Ldt$_{Mt2}$ S-pocket, Ldt$_{Mt1}$ has R25, Ldt$_{Mt3}$ has Q66, and Ldt$_{Mt5}$ has H219, and each of these putative S-pocket amino acids have similar basic charge properties (*Figure 7—figure supplement 1*). Moreover, superposition of the crystal structure of these paralogs with Ldt$_{Mt2}$–sugar complex places the PG sugar moiety within the S-pocket. We suggest a common S-pocket-mediated allosteric mechanism in all of the L,D-transpeptidases in *M.tb*; however,

the rate of transpeptidation may differ depending upon structural differences in their respective YkuD flaps, S-pockets, and catalytic sites.

Based on our crystal structure and modelling studies, we propose that prior to the 3–3 transpeptidation between the donor and acceptor PG stem peptides, the acceptor PG sugar moiety chain is anchored across the IgD1-YkuD domains interface to the S-pocket of Ldt$_{Mt2}$. PG sugar chain anchoring has been observed in L,D-transpeptidase of *B. subtilis* through a PG recognition domain LysM (*Schanda et al., 2014*). The LysM domain binds a sugar moiety of the PG precursor, and the tetrapeptide branch (acceptor stem) contacts the catalytic cysteine residue through the inner cavity of the catalytic domain. Another PG-binding enzyme lysostaphin from *Staphylococcus simulans* has a PG anchoring domain, SH3b, while its catalytic domain cleaves PG stem cross-bridge (*Mitkowski et al., 2019*). Upon anchoring of the PG sugar moiety chain within the S-pocket in Ldt$_{Mt2}$, its acceptor stem peptide binds to the inner pocket of the enzyme's catalytic domain, and the donor stem binds to the outer cavity close to the C354 residue, interactions that foster formation of the 3–3 transpeptide linkage (*Bianchet et al., 2017*; *Erdemli et al., 2012*; *Fakhar et al., 2017*). We hypothesize that, prior to 3–3 transpeptide linkage, both S-pocket and catalytic site may work in cooperativity to facilitate synchronous binding of two PG substrates (donor and acceptor substrates); however, this requires experimental validation using nascent PG substrates that are beyond the scope of our study. Nevertheless, in support of our proposed model, we have tested our cooperativity hypothesis on ß-lactam binding and hydrolysis activity in Ldt$_{Mt2}$ and found results that support the model.

Ldt$_{Mt2}$ plays a major role in the resistance of *M.tb* to β-lactam class of drugs (*Cordillot et al., 2013*; *Dubée et al., 2012*; *Gupta et al., 2010*; *Lavollay et al., 2008*; *Mainardi et al., 2005*). Among the β-lactam class of drugs, penicillins and cephalosporins are readily hydrolysed by this enzyme, while carbapenems are potent Ldt$_{Mt2}$ inhibitors (*Kumar et al., 2017*). Our findings reveal the role of both the S-pocket and the catalytic site in regulating β-lactam hydrolysis and inhibition by the carbapenem subclass. We demonstrate an allosteric cooperativity between the S-pocket and the catalytic site in the dual recognition of carbapenem drugs, with the former one binding the carbapenem drug non-covalently with a saturable binding and the latter one covalently through irreversible acylation of C354. A similar β-lactam-binding mechanism has been observed in penicillin-binding protein 2a (PBP2a) from *Streptococcus aureus* where one molecule of β-lactam occupies an allosteric site (with a saturable binding behaviour) 60 Å away culminating into the allosteric conformational changes in PBP2a with the opening of the active site and covalent binding with a β-lactam molecule (*Otero et al., 2013*). During the β-lactam-binding process, Ldt$_{Mt2}$ occupies one molecule of β-lactam at a reversible binding site (the S-pocket) 21 Å away from the catalytic site, and this interaction stimulates allosteric conformational changes across the YkuD catalytic flap and catalytic site to drive acylation by a second ß-lactam molecule in the catalytic pocket. We find the role of catalytic site equally important in stimulating reversible binding of β-lactam in the S-pocket. Thus, there we observe bidirectional cooperativity between the S-pocket and the catalytic site in binding dual β-lactams, and the same mechanism may apply to dual PG substrate binding. The role of differential dynamics by the YkuD flap in β-lactam- and substrate binding by the catalytic site have been demonstrated earlier by MD simulations (*Fakhar et al., 2017*); however, our study finds additional roles of YkuD flap in dual β-lactam binding in S-pocket and the catalytic site. Several allosteric alterations mediated by new water-mediated salt bridges or breakage of pre-existing ionic interactions contribute to the cumulative dynamics of the YkuD flap during dual β-lactam binding in Ldt$_{Mt2}$ (*Figure 6*, *Figure 3—figure supplement 2*). Also, through network analysis, we identify the pathways of allosteric communications that flow from the S-pocket via YkuD flap and directly to the catalytic site via a hydrophobic core to modulate the β-lactam binding (*Figure 3C* and *Figure 3—figure supplement 3*). The role of hydrophobic core residues in channelling allosteric communications to couple with catalysis has been observed in a β-lactam hydrolyzing class A β-lactamase (*Olehnovics et al., 2021*). Our study has also led to identification of a new residue S351 in the catalytic site that can be modulated by allostery (*Figure 3*, *Figure 3—figure supplement 2*). S351 can be an important part of the catalytic center in addition to the C354–H336–S337 catalytic triad. More investigational studies will be required in future to fully understand the role of S351 residue in catalytic function of Ldt$_{Mt2}$.

We find that various β-lactams bind to the S-pocket of Ldt$_{Mt2}$ through their tail regions, either through their R1 or R3 groups. As we found the docking scores of ampicillin, oxacillin, and cefotaxime to be higher than those of carbapenems, biapenem, and T203, the interactions of these R1 or R3

groups with the S-pocket appears to play critical role in the initiation of S-pocket binding, irrespective of fate of β-lactams in catalytic site. This discovery of a novel mechanism of β-lactam binding in $Ldt_{Mt2}$ reveals important new parameters in the development of new β-lactams for *M. tb*, and highlights the importance of the respective R1 and R3 side chains to both occupy the S-pocket and modulate strong inhibition at the catalytic site. This study provides a high-resolution crystal structure of $Ldt_{Mt2}$ with an experimental carbapenem T203 that offers the basis for cooperative binding of dual β-lactams in the S-pocket and catalytic site (*Videos 1 and 2*). Insights gained from this analysis may guide the acquisition and/or the design and synthesis of new inhibitors.

# Materials and methods

## Key resources table

| Reagent type (species) or resource | Designation | Source or reference | Identifiers | Additional information |
|---|---|---|---|---|
| Gene (*Mycobacterium tuberculosis*) | ldtB/LdtMt2 | Uniprot | I6Y9J2 | |
| Strain, strain background (*Escherichia coli*) | BL21 (DE3) | NEB | Catalog # C2526H | |
| Strain, strain background (*Escherichia coli*) | NEB 5-alpha competent | NEB | Catalog # C2987H | |
| Sequence-based reagent | IgD1-F | This paper | PCR primer | ATTGCCATATGAAGGCACGCCGTTCGCCGAC |
| Sequence-based reagent | IgD1-R | This paper | PCR primer | CAATACTCGAGTTAGGTCTGGAAGGTCAGCTGGCG |
| Sequence-based reagent | IgD2-F | This paper | PCR primer | ATTGCCATATGACCTGACCATGCCCTACGTAT |
| Sequence-based reagent | IgD2-R | This paper | PCR primer | CAATACTCGAGTTAGCCGATGGTGAAGTGCGTCTG |
| Sequence-based reagent | IgD1-IgD2-F | This paper | PCR primer | ATTGCCATATGAAGGCACGCCGTTCGCCGATC |
| Sequence-based reagent | IgD1-IgD2-R | This paper | PCR primer | CAATACTCGAGTTAGCCGATGGTGAAGTGCGTCTG |
| Sequence-based reagent | YkuD-F | This paper | PCR primer | ATTGCCATATGGGCGACGAGGTGATCGCGACC |
| Sequence-based reagent | YkuD- R | This paper | PCR primer | CAATACTCGAGTTACGCCTTGGCGTTACCGGC |
| Sequence-based reagent | $Ldt_{Mt2}$-E207A- F | This paper | PCR primer | CTGAATAACCGTGCAGTGCGTTGGCGCCCA |
| Sequence-based reagent | $Ldt_{Mt2}$-E207A- R | This paper | PCR primer | TGGGCGCCAACGCACTGCACGGTTATTCAG |
| Sequence-based reagent | $Ldt_{Mt2}$-R209A- F | This paper | PCR primer | TGAATAACCGTGAAGTGGAATGGCGCCCAGAGCATT |
| Sequence-based reagent | $Ldt_{Mt2}$-R209E- R | This paper | PCR primer | AATGCTCTGGGCGCCATTCCACTTCACGGTTATTCA |
| Sequence-based reagent | $Ldt_{Mt2}$-S337A- F | This paper | PCR primer | GTGTCTTCGTGCACGCAGCGCCGTGGTCGG |
| Sequence-based reagent | $Ldt_{Mt2}$-S37A- R | This paper | PCR primer | CCGACCACGGCGCTGCGTGCACGAAGACAC |
| Sequence-based reagent | $Ldt_{Mt2}$-S351A- F | This paper | PCR primer | GGCCACACCAACACCGCCCATGGCTGCCTGAAC |
| Sequence-based reagent | $Ldt_{Mt2}$-S351A-R | This paper | PCR primer | GTTCAGGCAGCCATGGGCGGTGTTGGTGTGGCC |
| Sequence-based reagent | $Ldt_{Mt2}$-Y330F- F | This paper | PCR primer | CCACCCAGATCTCCTTTAGCGGTGTCTTCGTGC |
| Sequence-based reagent | $Ldt_{Mt2}$-Y330F- R | This paper | PCR primer | GCACGAAGACACCGCTAAAGGAGATCTGGGTGG |
| Sequence-based reagent | $Ldt_{Mt2}$-H336N- F | This paper | PCR primer | CAGCGGTGTCTTCGTGAA**C**TCAGCGCCGTGGTC |
| Sequence-based reagent | $Ldt_{Mt2}$-H336N- R | This paper | PCR primer | GACCACGGCGCTGAGTTCACGAAGACACCGCTG |
| Sequence-based reagent | $Ldt_{Mt2}$-C354A- F | This paper | PCR primer | CAACACCAGCCATGGCGCGCTGAACGTCAGCCCGAG |
| Sequence-based reagent | $Ldt_{Mt2}$-C354A- R | This paper | PCR primer | CTCGGGCTGACGTTCAGCGCGCCATGGCTGGTGTTG |
| Sequence-based reagent | $Ldt_{Mt2}$-M303A- F | This paper | PCR primer | GGTACAAGCACATCATCGCGGACTCGTCCACCTACG |
| Sequence-based reagent | $Ldt_{Mt2}$-M303A- R | This paper | PCR primer | CGTAGGTGGACGAGTCCGCGATGATGTGCTTGTACC |
| Commercial assay or kit | Q5 High-Fidelity DNA Polymerases | NEB | Catalog # M0491L | |
| Chemical compound, drug | SYPRO Orange | Thermo Fisher Scientific | Catalog # S6650 | |

*Continued on next page*

*Continued*

| Reagent type (species) or resource | Designation | Source or reference | Identifiers | Additional information |
|---|---|---|---|---|
| Chemical compound, drug | Nitrocefin | Millipore-Sigma | Catalog # 484400-5MG | |

## Cloning and site-directed mutagenesis

DNA sequences encoding $Ldt_{Mt2}$-Δ42, $Ldt_{Mt2}$-Δ55, IgD1 (50–145 aa residues), IgD2 (150–250 aa), IgD1–IgD2 (50–250 aa), and YkuD domain (250–408) and CTSD deletion mutant $Ldt_{Mt2}$ 42–384 were cloned in pET28a vector to express the protein with *N*-terminal $His_6$-tag that is cleavable by Tobacco Etch Virus (TEV) protease. Single amino acid substitutions of $Ldt_{Mt2}$-Δ55 were constructed by site-directed mutagenesis for the following mutations: E207A, R209E, Y330F, C354A, H352A, H336A, M303A, S337A, and S351A.

## Protein expression and purification

Mutants and different fragments of $Ldt_{Mt2}$ were expressed and purified as reported earlier (*Kumar et al., 2017*). In detail, $Ldt_{Mt2}$-ΔN55 was transformed in chemical competent *E. coli* BL21δε3 (NEB labs). A single colony of transformed cells was inoculated in 50 ml of Luria-Bertani (LB) media supplemented with ampicillin (100 µg/ml) before growing overnight (O/N) at 37°C in an incubator shaker. The O/N culture was used to inoculate secondary culture in LB media to grow at 37°C until the optical density at 600 nm reached ~0.6–0.8. At this stage, temperature was lowered to 16°C in the incubator shaker before inducing the protein expression with 0.5 mM of isopropyl-1-thio-β-galactoside. The secondary culture grown O/N. The culture was harvested and the cell pellet was resuspended in lysis buffer (50 mM Tris buffer pH 7.5, 400 mM NaCl, 10% glycerol, 1.0 mM dithiothreitol [DTT], and 1.0 mM phenyl-methylsulfonyl fluoride [PMSF]). 0.5 mg/ml lysozyme was added into the resuspended cells to allow cell lysis at 4°C for 30 min. Resuspended cells were further lysed by ultrasonication at 4°C with a pulse rate of 15 s ON/OFF. Whole cell lysate was centrifuged at 10,000 × *g* for 45 min and the supernatant was loaded onto Ni-NTA column (Qiagen, Germany). The unbound protein was washed with washing buffer (50 mM Tris buffer pH 7.5, 400 mM NaCl, 10% glycerol, 1.0 mM DTT, 0.1 mM PMSF) and the protein was eluted with elution buffer (50 mM Tris buffer pH 8.0, 400 mM NaCl, 1.0 mM DTT, 0.1 mM PMSF, and 500 mM imidazole). The $His_6$-tag of the protein was removed by TEV protease during overnight dialysis against the buffer 50 mM Tris pH 8.0, 150 mM NaCl, and 1.0 mM DTT at 4°C. The dialyzed protein was passed through Ni-NTA column and the $His_6$-tag-removed protein was collected in flow-through. Protein was further purified using superdex 10/300 column on ÄKTA pure 25. The purified protein was concentrated to 20 mg/ml as measured by nanodrop at 280 nm wavelength. The purity of protein was checked by 12% sodium dodecyl sulphate–polyacrylamide gel electrophoresis. All other truncation and mutants of $Ldt_{Mt2}$ were also purified by same protocol as above, however their $His_6$-tag was not removed.

## ThermoFluor assays

The proteins $Ldt_{Mt2}$-Δ55, R209E, and S351A were with initial stocks of 11.5, 14.0, and 21 µM, respectively, in the 50 mM Tris buffer pH 8.0, 150 mM NaCl, 1 mM DTT. 5000× of SYPRO Orange (Invitrogen) was diluted to 50× in water. 5 µM of proteins and 3× of SYPRO Orange were pipetted into a 96-well PCR plate (BioRad, MicroAmp Fast 96-Well Reaction plate, 0.1 ml) with 50 µl total volume in the well. Fluorescence data were collected on BioRad StepOnePlus Real-Time PCR System using the software StepOne software v2.3. ROX (SYPRO Orange) was selected as a reporter dye and none for passive reference in the software. The temperature was held for 1 min per degree from 25 to 65°C. Melting temperature ($T_m$) and differential fluorescence ($-dF/dT$) values were calculated by fitting the data on Sigmoidal dose–response (variable slope) equation in GraphPad Prism software. Experiments were performed in biological triplicates.

## Nitrocefin hydrolysis assays

Nitrocefin (Calbiochem) with a range of 1–400 µM was used as a substrate for quantifying the rate of β-lactam hydrolysis by different $ldt_{Mt2}$ fragments and mutants. A 100 µl reaction mixture containing 5 µM enzyme in 25 mM HEPES (hydroxyethyl piperazineethanesulfonic acid)–MES

(morpholinoethanesulfonic acid)–Tris-phosphate buffer, 300 mM NaCl, pH 6.0, was incubated at 25°C. Nitrocefin hydrolysis was measured at 496 nm on BioRad microplate reader and the absorbance data were converted to µM/minute using Beer's Law ($e$ = 20,500 $M^{-1}$ $cm^{-1}$ for hydrolysed nitrocefin; $L$ = 0.5 cm). Nitrocefin hydrolysis assays were performed in experimental duplicates and rate constants, $V_{max}$ and $K_m$ were calculated by fitting the data on nonlinear regression curve with Michaelis–Menten equation.

## CD spectroscopy

Far-UV CD spectra were acquired on a Jasco-815 spectropolarimeter. $Ldt_{Mt2}$ and R209E mutant proteins with a concentration of 5.0 µM were used in the study. Cuvette of path length of 0.2 cm was used and spectra were collected from 260 to 190 nm at a rate of 100 nm/min and data pitch of 1 nm, with averaging of 10 scans for noise reduction. Contribution of the buffer to the spectra was electronically subtracted and $\theta_{obs}$ was plotted.

## Biapenem acylation assays

The acylation of biapenem with $Ldt_{Mt2}$ and mutants was determined by measuring the reduction in absorbance of biapenem at 292 nm wavelength using UV–visible spectrophotometry. A 100 µl reaction mixture containing 50 µM enzyme, 50 µM biapenem, 25 mM Tris buffer pH 7.5 was incubated at 15°C and endpoint absorbance was recorded at 30-s intervals for 8 min. Rate constant ($K$) of biapenem acylation was calculated by fitting the data on a nonlinear regression curve with one-phase decay. Experiment was performed in biological triplicates to calculate standard deviation and average values.

## Protein crystallization

Purified $Ldt_{Mt2}$ (fragment ΔN55) was crystallized with the same conditions has reported earlier (*Kumar et al., 2017*). Crystals were grown by hanging drop vapor diffusion method in 20% 5000 MME and 200 mM ammonium sulphate condition. For $Ldt_{Mt2}$–T203 complex, crystals were soaked with 2 mM of T203 drug overnight before being cryo-protected in 20% 5000 MME, 30% glycerol, and 120 mM ammonium sulphate before flash freezing in liquid nitrogen.

## Crystal diffraction, data collection, and structure determination

The crystals were diffracted at 100 K temperature at a wavelength of 1.0 Å on beamline 19-ID at the Advanced Photon Source (Argonne National Laboratory). The diffraction data were recorded on an ADSC Quantum 315r CCD detector and processed with the HKL3000 software (*Minor et al., 2006*). The crystal structures of $Ldt_{Mt2}$–sugar complex at a highest resolution of 1.58 Å and $Ldt_{Mt2}$–T203 complex at 1.7 Å resolution were solved by molecular replacement method using *PHENIX* suite of program (*Liebschner et al., 2019*) using the coordinates of $Ldt_{Mt2}$ (PDB ID: 5DU7) as a search model. The initial structures were subjected to crystallographic refinement with *phenix.refine* (*Afonine et al., 2012*) from the *PHENIX* suite of programs. Structures were rebuilt with COOT (*Emsley and Cowtan, 2004*) to fit the electron density map. Structure validation was done using Molprobity (*Williams et al., 2018*). The $R$ values of refined structures (*Table 1*) are well within the range of typical resolution. Omit maps for ligands in the structures were created from map coefficient using *PHENIX* suite of programs. Figures were prepared using PyMOL Molecular Graphics System, Version 2.4.0 Schrödinger, LLC.

## Docking studies

The structural model of $Ldt_{Mt2}$ was prepared in Autodock Tools 1.5.7 (*Trott and Olson, 2010*) by adding hydrogen atoms, noBondOrder method and Gasteiger charges of −13.019. Ligands (namely ampicillin, cefotaxime, oxacillin, biapenem, and T203) were prepared having five rotatable bonds. A grid was assigned with a grid box size 60 Å × 60 Å × 60 Å and grid spacing 0.375 Å. Docking was performed using Genetic Algorithm with number of runs set to 100, a population size of 50, maximum number of evaluations on medium as 1,500,000 and all other default parameters on Autodock4 program.

## MD simulations setup

MD simulations were performed for (1) crystal structure of $Ldt_{Mt2}$ docked with biapenem only at S-pocket ($Ldt_{Mt2}$–$Bia^S$), (2) crystal structure of $Ldt_{Mt2}$ docked with Biapenem only at catalytic site ($Ldt_{Mt2}$–$Bia^C$), and

(3) crystal structure of $Ldt_{Mt2}$ docked with biapenem at S-pocket and at catalytic site ($Ldt_{Mt2}$–$Bia^{S-C}$). Explicit water (TIP3P) MD simulations of $Ldt_{Mt2}$–biapenem docked complexes were carried out with AMBER16 employing *ff03* forcefield (*Duan et al., 2003*) and general AMBER forcefield (*gaff*) (*Wang et al., 2004*) for protein and ligands, respectively. The Leap module of AMBER16 was used for setting up initial structures. To prevent any steric clashes between solute and solvent during MD simulation, all the solvated systems were initially minimized in two steps. First, minimization of solvent and ions was performed by applying 50 kcal/mol/$Å^2$ positional restraint on all the atoms of protein and ligand, followed by the second minimization of the whole system without any positional restrain. The system was then heated using Langevin dynamics from 10 to 300 K at NVT ensemble with the positional restraint of 5 kcal/mol/ $Å^2$ on the protein and ligands heavy atoms. Positional restraint was gradually released in the next two steps, that is, 3 kcal/mol/$Å^2$ in first step and then 1 kcal/mol/$Å^2$ in second step. Further, the system was equilibrated for 100 ps at NVT followed by 2400 ps at NPT ensemble. Finally, the production runs ($Ldt_{Mt2}$–$Bia^S$ = 75 ns, $Ldt_{Mt2}$–$Bia^C$ = 75 ns, and $Ldt_{Mt2}$–$Bia^{S-C}$ = 75 ns) were carried out at NPT ensemble by integrating the Newtonian equation of motion at every 2 fs. Trajectories were analyzed using cpptraj module of AMBER16 (*Roe and Cheatham, 2013*). Trajectories were analyzed by cpptraj module of AMBER16. Hydrogen bonds were calculated for the donor–acceptor distance cutoff of 3.5 Å and the donor–hydrogen–acceptor angle cutoff of 135°.

Additional MD simulations were performed with (1) $Ldt_{Mt2}$ (wild-type); (2) $Ldt_{Mt2}$-R209E; (3) crystal structure $Ldt_{Mt2}$–$T203^{S-C}$ with T203 drug in both S-pocket and catalytic site; and (4) crystal structure $Ldt_{Mt2}$–$T203^S$ with T203 drug in S-pocket alone. IgD1 domain was cleaved from all four systems using BIOVIA's Discovery Studio 2021 software (Dassault Systems). All MD simulations were performed with NAMD 2.14 software using CHARMM36 forcefield (*Phillips et al., 2020*). Systems were first prepared using CHARMM program and then solvated in a water box of minimum distance 15 Å from any edge in VMD. Neutralization was performed by addition of $Na^+$ and $Cl^-$ ions. Stepwise energy minimization performed using the Steepest Descent algorithm for 500,000 steps. The system was heated from 0 to 300 K in 10 K/ps increments with the NAMD program (*Phillips et al., 2020*). Velocities further were rescaled for the next 200 ps, followed by 2000 ps of unconstrained equilibration. Subsequently, the production runs of 200 ns each was performed for the $Ldt_{Mt2}$ wild-type and R209E mutant protein. Production run for the $Ldt_{Mt2}$–$T203^{S-C}$ that is in complex with dual T203 drug was performed for 50 ns. Production was not performed for $Ldt_{Mt2}$–$T203^S$ that in complex with T203 drug only at S-pocket, since T203 drug left the S-pocket during equlibration 2 at 650 ps. All simulations were performed in the NPT ensemble using the periodic boundary conditions. Water was represented by the TIP3P model. A 12 Å cutoff was used for nonbonded interactions and long-range electrostatic interactions were treated with the smooth PME method as implemented in NAMD. All bonds to hydrogen atoms were constrained allowing for a timestep of 2 fs. Analysis was performed using standard GROMACS (*Pronk et al., 2013*), Visual Molecular Dynamics (VMD) tools (*Humphrey et al., 1996*). Probability Density Graphs were plotted using R Programming in RStudio.

### Clustering of conformers and network construction

Clustering was performed for each simulation using gromos method from the GROMACS software and a representative structure from the largest cluster was considered to average the distance calculations and also to construct a network of amino acid residues. Network was calculated and constructed using the NAPS webs server (Network analysis of Protein Structures) (*Chakrabarty and Parekh, 2016*) and visualized using Cytoscape. For network construction each amino acid was considered as a node in the network and an edge was constructed if the distance between a pair of atoms of the residue pair was within the lower threshold of 0 Å and upper threshold of 5 Å. All edges were considered unweighted (equally important). Degree of nodes, node betweenness, edge betweenness, clustering coefficients, closeness, eigen vector centrality, eccentricity, and average neighbour degree were computed as an estimate of centrality. Normalized covariance (correlation) of MD simulation was performed using CARMA (*Glykos, 2006*). The degree of coupled motion in $Ldt_{Mt2}$ was measured by normalizing the cross-correlation matrix of atomic fluctuations over the length of the simulation.

### Acknowledgements

T203, an experimental carbapenem, was a kind gift from Dr. Joel S Freundlich at Rutgers University Medical School. We are grateful to Prof Tomasz Borowski at Jerzy Haber Institute of Catalysis and

Surface Chemistry, Prof Craig Townsend and Dr. C Korin Bullen at Johns Hopkins University. Experimental support from Pallavi Juneja at Jamia Hamdard in media and reagent preparation is highly appreciated. We thank personnel at Argonne National Laboratory for data collection of protein crystals. This research was supported in part by PL-Grid Infrastructure. Computations were performed at Academic Computer Centre Cyfronet AGH. A part of this research was also conducted using computational resources (and/or scientific computing services) at the Maryland Advanced Research Computing Center (MARCC).

## Additional information

### Funding

| Funder | Grant reference number | Author |
|---|---|---|
| Science and Engineering Research Board | CRG/2019/005079 | Pankaj Kumar |
| National Institutes of Health | R33 AI111739 | Gyanu Lamichhane |
| Department of Biotechnology, Ministry of Science and Technology, India | BT-RLF/Re-entry/68/2017 | Pankaj Kumar |
| National Institutes of Health | R01 AI155664 | Gyanu Lamichhane |

The funders had no role in study design, data collection, and interpretation, or the decision to submit the work for publication.

### Author contributions

Nazia Ahmad, Methodology, Validation, Writing – original draft; Sanmati Dugad, Conceptualization, Methodology, Validation, Visualization, Writing – original draft; Varsha Chauhan, Shubbir Ahmed, Kunal Sharma, Sangita Kachhap, Methodology; Rana Zaidi, Resources, Supervision; William R Bishai, Methodology, Writing – original draft, Writing - review and editing; Gyanu Lamichhane, Conceptualization, Funding acquisition, Investigation, Methodology, Resources, Supervision, Validation, Writing – original draft; Pankaj Kumar, Conceptualization, Funding acquisition, Investigation, Methodology, Project administration, Supervision, Validation, Writing – original draft, Writing - review and editing

### Author ORCIDs

William R Bishai http://orcid.org/0000-0002-8734-4118
Gyanu Lamichhane http://orcid.org/0000-0002-2214-0114
Pankaj Kumar http://orcid.org/0000-0001-9163-3273

### Decision letter and Author response

Decision letter https://doi.org/10.7554/eLife.73055.sa1
Author response https://doi.org/10.7554/eLife.73055.sa2

## Additional files

### Supplementary files
• Transparent reporting form

### Data availability
Diffraction data have been deposited in PDB under the accession code 7F71, 7F8P.

The following datasets were generated:

| Author(s) | Year | Dataset title | Dataset URL | Database and Identifier |
|---|---|---|---|---|
| Kumar P, Lamichhane G | 2021 | Crystal structure of the Mycobacterium tuberculosis L,D-transpeptidase-2 (LdtMt2) with peptidoglycan sugar moiety and glutamate | https://www.rcsb.org/structure/unreleased/7F71 | RCSB Protein Data Bank, 7F71 |
| Kumar P, Lamichhane G | 2021 | Crystal structure of the Mycobacterium tuberculosis L,D-transpeptidase-2 (LdtMt2) with new carbapenem drug T203 | https://www.rcsb.org/structure/unreleased/7F8P | RCSB Protein Data Bank, 7F8P |

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
