## [Editor Report]

This manuscript reports high-resolution crystallographic structures of the L,D-transpeptidase from *Mycobacterium tuberculosis*, which was obtained with ligands (a sugar molecule and a β-lactam). A surprising finding is that the enzyme contains a ligand-binding site located greater than 20 Å away from the catalytic site. The authors used mutagenesis studies and computational analyses to support an allosteric role for the new ligand site (S-pocket). This site could potentially permit the inhibition of L,D-transpeptidases.

---

## [Decision Letter]

**Decision letter after peer review:**

Thank you for submitting your article "Allosteric cooperation in ß-lactam binding to a non-classical transpeptidase" for consideration by *eLife*. Your article has been reviewed by 3 peer reviewers, and the evaluation has been overseen by a Reviewing Editor and Olga Boudker as the Senior Editor. The following individual involved in review of your submission has agreed to reveal their identity: Shozeb Haider (Reviewer #1).

Essential revisions:

1) Please include details of the molecular dynamics simulations so that reviewers can assess their rigor. 2 reviewers mentioned that the 100 ns trajectories used in the simulations might not be sufficient to reveal real conformational dynamics. Please address by increasing the trajectories.

2) Please provide parameters used for the docking studies.

3) Please present binding data for β-lactam binding to the S-pocket in protein with active-site mutations.

4) Please provide additional data investigating the importance of other residues (E168, R371, Y330 and A171, M157 and L391) in the S-pocket to bolster the data with R209.

*Reviewer #1 (Recommendations for the authors):*

The structures of LdtMt2 have been published before, both as apo and in complex with a β-lactam molecule. The current structures report on a novel allosteric site that was not previously described.

Figure 2A should be moved to Figure 1.

The structures in Figure 1C (purple and green) that are superimposed do not look superimposed. No quantitative measure (RMSD) is provided. What parts of the structure were superimposed? Perhaps a sequence alignment between LdtBS and LdtMt2 might explain this better. Further qualitative analysis of the inner cavity should be carried out. The electrostatic surface should be shown for both proteins and how the corresponding electrostatic surface of the PG complements it should be illustrated.

Dynamic distances between important residues (H336-C354, H336-S351, H336-S337, etc.) over the course of the simulation should be plotted to support the claims that the authors make.

One of the strengths of this paper is the finding that S351 and not S337 might form the catalytic triad. The authors should strengthen this by quantifying structural interactions that support mutagenesis experiments. A wealth of data is present in the MD simulations; they should use this to their advantage.

Since there are other structures of mutants of LdtMt2 present, why did the authors not try to crystallize R290E and other mutants? The structural findings would have been definitive in assessing the structural differences between the apo and ligand-bound states. A comment should be made in the discussion.

The manuscript relies heavily on MD simulations to structurally explain a number of experimental observations. The computational study is an essential part of the manuscript, yet it seems to be carried out like an afterthought. MD simulations analysis is poorly carried out. The details in the method section for both automated docking and MD simulations are poor and lack details to reproduce the study. For example, How were the ligands parameterized? No evidence has been provided which even remotely suggests that the simulated system has been equilibrated, let alone converged. How many simulations were run to confirm the observations were statistically significant? The interpretations of the results are equally perplexing. The tumbling of the ligand in the binding site is because the system has not equilibrated and should be omitted from being included in any interpretation (line 334-335). 100ns might be too short for the essential interactions to stabilize. If the authors think otherwise, they should support that with evidence. None of that has been included. MD simulation figures in the supplementary are poor and do not provide any information that strengthens the excellent biochemical analysis and mutagenesis experiments. The MD simulation figures in Supplementary are not even labeled appropriately.

Why have the dynamics of R209 not been discussed? If the allosteric signals are indeed emanating from this residue (or any in the allosteric pocket), then surely a direct link could be traced between R209 (or other residues) and the catalytic triad? A network analysis should be able to resolve this. Or some equivalent method that demonstrates this structural link.

Line 342/344/346: There is no Figure S4A or S4B. Just S4

Line 347-350: "Thus, MD simulations suggest that biapenem binding across the S-pocket surface imposes stability in fluctuations of β-lactam movement in the catalytic site and the.-lactam ring carbonyl group maintains a close distance with the Sϒ atom of C354 that favor a nucleophilic attack"

It might very well be that this is indeed the case. But without evidence from MD simulation, it seems like they are an over-interpretation of results.

The docking protocol was baffling. When the authors have a crystal structure with a ligand, then what was the reason they did not carry out a template-based docking? Surely, the β-lactam ring in T203 could have been used as a fixed focal point. Yet, they even dock T203. If the authors were trying to do a blind dock to confirm the binding mode of T203 (because of the poor electron density), then they should say so and also show the results of the dock, superimposed on the crystal structure. No such thing has been detailed in the method or discussion.

The authors mention that the electron density of T203 is poor. Therefore in addition to 2Fo-Fc maps, they should also include omit maps. This will only strengthen their claim of how they have modeled the ligand in the sites.

Line 449: What was this distance measured between? Which residues or atoms?

Line 452-453: "Mutational changes in the S-pocket or the catalytic pocket nullifies all the allosteric communications that are otherwise important in the cooperative binding of dual ß-lactams." How is this structurally achieved between the two sites that are far apart? The authors should use network analysis to resolve this?

Line 471-472: There is surely a mechanism of allosteric communication, its just that the authors have not provided evidence for that. What the authors have shown is the end states in either the allosteric site or the catalytic site, but not how signals would be communicated upon ligand binding in the allosteric site to the catalytic site.

Line 479: Why have the properties (structural or sequence) have not been shown for the putative S-pocket. A close-up (structural superimposition of conserved residues) or sequence alignment should have been presented. Perhaps figures of electrostatic charge surface would convey the point the authors are trying to make. Figure 8 is the least informative as it currently stands.

*Reviewer #2 (Recommendations for the authors):*

In this work, authors seek a better understanding of the L,D-transpeptidase class of enzymes. They investigate Ldt-Mt2, 1 of five L,D-transpeptidase paralogs rom *Mycobacterium tuberculosis*. First, they determine a crystal structure of LdtMt2 and are able to model density for glucose into a pocket between two domains YkuD and IgD2. This pocket is hereafter referred to as the S-pocket. Authors use Thermofluor to provide support for glucose binding at that (using a peptidoglycan/PG precursor) site. Mutation of a residue in the S-pocket: R209E disrupts the binding of the precursor. By deleting selected domains, authors are able to show that both domains YkuD and IgD2 are necessary for hydrolysis activity, supporting a role for the S-pocket in governing catalytic activity. MD simulations were used to investigate the influence of S-pocket mutations on dynamics at the active site (21 angstroms away). Distinct fluctuations in active site residues were observed when the S-pocket mutation was introduced in the simulations. Upon mutation, these active site residues affected β-lactam hydrolysis, indicating that fluctuations observed in MD were functionally relevant.

The S-pocket mutation was shown to disrupt irreversible acetylation by an inhibitor (biapenem) at the active site. Thermoflour showed that biapenem likely binds Ldt-Mt at two sites, covalently at one and non-covalently at the other. Both the S-pocket mutation and an active site catalytic mutant hindered both binding events, supporting an interplay between the two sites. MD simulations and docking are used to further probe cooperativity between the two sites. When biapenem is bound at both sites, increased stability is observed in the molecule compared to when biapenem is bound to only one site.

Finally, authors dock other classes of β lactams into Ldt-Mt2 and observe similar binding modes for all. Thermoflour shows that the R209E mutation disrupts binding of these β lactams. A second crystal structure of Ldt-Mt2 is presented with T203 drug. T203 densities were modeled into the S-pcoket and catalytic site. Structural comparison to other LdtMt2 structures revealed alterations in both S-pocket and catalytic site regions. In summary, authors conclude that the S-pocket is an allosteric site that controls catalytic activity of Ldt-Mt2.

This study is potentially very exciting and very important, as the report of this allosteric site is novel and might be a prevailing mechanism across other L,D-transpeptidases.

There are two major issues that I believe should be addressed. (1) Because of the nature of the evidence presented to support the existence/role of the S-pocket (i.e. computational modeling, ligands modeled into electron density, a single disruptive mutation), it is my opinion that the authors have to provide an additional piece of data to support the existence of the S-pocket. (2) The computational studies presented, in my opinion, are not rigorous enough to support the claims that are made. My comments are provided in detail below.

1. After modeling glucose into the S-pocket, authors propose R209, E168, R371, Y330 and A171, M157 and L391 as residues interacting with the sugar. However only a single mutation (R209E) disrupts binding of the PG precursor. The identification of only a single mutation is strange, because for a bonafide binding site, there should be at least > 1 residue which can disrupt binding. Because other mutations are not reported that disrupt binding at S-pocket, there is not strong evidence that the S-pocket is an actual site for binding ligands. This poses an important question: is it possible that R209E affects Ldt-Mt2 function by just disrupting the IgD2-YkuD domain interface and subsequently destabilizes the active site? To confirm that the S-pocket is actually a distinct site that binds ligand and allosterically regulates the active site, there needs to be evidence that R209E does not merely disrupt dimerization of the two domains. The majority of the case for the existence of this pocket hinges on functional studies with R209E, therefore this possibility should be ruled out.

2. Authors have extracted quite a bit of mechanistic detail from their simulations, including demonstrating cooperativity between the two sites. However authors should be cautious, because 100 ns trajectories are not long enough to give meaningful information about fluctuations or persistence of conformations. For a protein with 450 residues, my sense is that the trajectories are barely equilibrated at 100 ns. However RMSDs over the trajectory can be shown to give some indication of how overall structure is changing between 0 and 100 ns. This will help the authors determine whether the changes in active site residues that they report can be expected to persist over longer timescales. One suggestion to achieve longer trajectories: authors could truncate the complex to just the two domains of interest, which might allow increased simulation time to further support these claims.

This is a potentially exciting piece of work. To make this suitable for publication, authors must provide a little more evidence that the S-pocket is a site that binds ligand. For both ligands modeled into electron density, the observation of only small fragments of these ligands casts doubt on the veracity of their binding. Likewise, the identification of only a single mutation that disrupts S-pocket is not sufficient to support binding.

Additionally, for some of the claims being made using docking and MD simulations, particularly those of cooperativity and pocket cross-talk, authors have not demonstrated that their trajectories are long enough to extract these conclusions. Evidence should be provided to test equilibration and confirm that the key interactions/conformational changes presented here will persist in longer simulations. Alternatively, authors could add additional experimental validations to demonstrate cooperativity between the two sites.

*Reviewer #3 (Recommendations for the authors):*

Abstract: the first sentence suggests that only *M. tuberculosis* has Ld transpeptidases, which is incorrect. Please edit this accordingly. The site termed 'S-pocket' on Line 42 is referred to as 'S-site' on line 45; please adopt a consistent terminology.

Introduction: although it is well-written, I do not think the summary of the paper (Lines 149-155) accurately reflect the results of the paper. In my view, the results on β-lactams and not peptidoglycan are the significant portion of the paper. Also, claims such as 'several interdisciplinary approaches' and 'explain the mechanism for the manifestation of activity of Ldtm2' are not completely supported, since the authors uncover an additional facet of the functioning of the enzyme. These sentences should be tempered.

Line 161: it is not clear whether the crystal structure at 1.57A is an improvement on previous efforts and whether this improved resolution is what enables the authors to discover the glucose binding in the S-pocket. The text should be re-written to make these points clear.

Figure 1B: comparable data for glucose would help the readers assess the strength of the conclusions regarding PG. In this panel, and in similar panels in other figures, it is not clear if the trend is significant. Could the authors show error bars for such figures, and perhaps include a plot of Tm vs. substrate concentration in the inset to highlight the trend better?

Figure 1D: the inner and outer cavity in the catalytic site that is discussed in the text is not labelled in the figure, and so it is hard to appreciate the location of the larger pentameric PG molecule. Furthermore, the authors could show Thermofluor data for a suitable point-mutant in the inner cavity to confirm their claims that the PG molecule can be located within the S-pocket and the inner cavity and complement the data in Figure 1B.

Lines 200-205: it is unclear what the phrases 'deletion of IgD2 from the YkUD domain as assessed from the YkuD domain alone' means. At Line 205, the S-pocket is described as being carried by the IgD2 domain but

Lines 271-273: while I do not disagree qualitatively with the descriptions, there is clearly a large shift in Tm at the lowest concentration of bioapenem, which is not reflected in the text.

Line 304: in contrast, the decreased stability of the S351A mutant has nothing to do with the biapenem binding – the statement describing this is misleading, Please revise this.

Figure 4B: based on the statements in the text, the effect of biapenem binding on melting structure is completely abroagated by the R209E and S351A mutations, yet the graphs show red arrows for the change in Tm. Please remove these if the trend is non-significant.

Figure 4C: given the content-rich format of *eLife*, the authors should provide animations of the simulations they describe, which would be far more informative than the snapshots presented.

Figure presentation: in many figures, the red arrow labels are hard to see. Please make the arrows bigger.

Figure 6 and last section of the Results: the motivation for studying in detail the molecule T203 is unclear. Is it because it has the most dramatic effects based on the results of Figure 5? The authors should cite relevant publications for this drug and include in the discussion how such a finding on an experimental carbapenem may be beneficial, as this is a strength of this manuscript.

Figures and figure legends: many methodological details e.g., the software used for the curve fitting can be moved to the Materials and methods and do not need to be repeated for every figure.

[Editors’ note: further revisions were suggested prior to acceptance, as described below.]

Thank you for resubmitting your work entitled "Allosteric cooperation in ß-lactam binding to a non-classical transpeptidase" for further consideration by *eLife*. Your revised article has been evaluated by Olga Boudker (Senior Editor) and a Reviewing Editor.

The manuscript has been improved but there are some remaining issues that need to be addressed, as outlined below:

1) S-pocket: R209 continues to be the only residue with a significant impact on ligand binding/catalysis. Y330 was used as a second proxy for the S-site, but its effect is much more muted than R209 or E207. The data with E207A, while not comprehensive (e.g. nitrocefin hydrolysis), does support the S-site more than data with Y330F – please note this.

2) The CD spectra of wt vs R209A show some differences ~ 210 nm that could be explained by reduced β-sheet content in the spectrum of the variant protein. Such a change could indicate problems with the protein fold that might account for the differences in catalytic activity. Can the spectra be fit to determine whether this difference is meaningful? This will bolster the data suggesting that R209 forms part of an allosteric site.

3) Cooperativity between the catalytic site and the active site: there is no clear evidence that cabapenam binding occurs cooperatively. In the absence of a quantitative assessment of cooperativity, it might be best to state that this is a possibility rather than a phenomenon that has been determined.

4) The binding of T203 to the active site is clear, but the 2Fo-Fc map shows low occupancy at the S-site. Were greater concentrations of the drug used? IF so, do those data reveal improved occupancy at the S-site?

Additionally, the comments below were provided by a reviewer.

*Reviewer #2 (Recommendations for the authors):*

Ahmad et al., have addressed my largest concern, going to extra steps to make sure that other S-pocket mutations disrupt small molecule binding. This portion of the manuscript is more convincing and the discovery of the S-pocket is quite novel.

While simulations have been increased, the authors still did not address the question of equilibration or show RMSDs over the length of the trajectory to show the feasibility of extracting mechanistic details from these simulations. However, the description of the conformational differences between wildtype and mutant Ldt-Mt2 is quite compelling.

Overall, concerns still remain about the computational work.

1. Authors mention that they have computed a dynamic network analysis for their MD trajectories. There is no explanation of what this analysis means or what information it provides. The method section is not instructive about the nature of this analysis, except to say that it was performed using VMD. It is also not my understanding that this method is used to calculate 'distances', as these can be calculated directly from the trajectory. To my knowledge, dynamic network analyses should minimally include discussions of one or more of the following: nodes and/or edges, suboptimal communication pathways, community structures, correlations, etc. If this detail about dynamic network analysis remains, there needs to be an explanation of what the approach is (explicit details in the Methods section), what information was generated for their trajectories and if possible, figures showing exactly what was extracted. As it stands now, all that is present are these distance plots and the density representations, which are informative but not dynamic networks. See https://doi.org/10.1073/pnas.0810961106 as a reference.

2. The docking study with biapenem is a weakness of this work. Because this part relies on docked and not crystal structures, the premise is tenuous to begin with. The fact that the small molecules fly out by 40 ns even in the best case is not reassuring that there is real binding. The most convincing way to use MD to make this point would be to use the crystal structure with the T203 at both sites as a starting point. My strong recommendation would be to remove this portion of the study or strengthen the work significantly by repeating the study with T203 and seeing whether the previously observed cooperativity persists.

---

## [Author Response]

Essential revisions:1) Please include details of the molecular dynamics simulations so that reviewers can assess their rigor. 2 reviewers mentioned that the 100 ns trajectories used in the simulations might not be sufficient to reveal real conformational dynamics. Please address by increasing the trajectories.

As per the suggestions, an additional 200 ns of MD simulations have been performed along with a dynamic network analysis of selected pair of catalytic site residues in the YkuD domain to consolidate the results with the experimental data. The details of the MD simulations have also been added in the Materials and methods.

– New Figure 3A contains a snapshot of trajectory at 200 ns for wild-type and mutant.

– New Figure 3B and Figure 3—figure supplement 1 show distance network analysis of selected pair of residues for 200 ns of MD simulations.

– We have also provided new Figure 3-source data 1 and Figure 3-source data 2 to support new Figure 3.

2) Please provide parameters used for the docking studies.

We have provided parameters that were used in the docking studies.

3) Please present binding data for β-lactam binding to the S-pocket in protein with active-site mutations.

Additional mutations have been performed in active-site residues as well as S-pocket to strengthen the data on f ß-lactam binding to the S-pocket.

– New Figure 4B along with source data 5 has been added that represent data on the rate of ß-lactam acylation (K_obs_ [50µM]/s^-1^) with various S-pocket and active-site mutants.

– New Figure 4C shows a graph with change in melting temperature (∆Tm) verses increasing concentration of ß-lactam drug. The data represents ß-lactam binding studies with several S-pocket (R209E, E207A, Y330F) and active-site mutants (S351A, H336N and C354A). A new Figure 4 – source data 1 supports Figure 4C.

– New Figure 4D contains ThermoFluor data on differential fluorescence change (-dF/dT) verses ß-lactam drug concentrations. A new Figure 4 – source data 1 supports Figure 4D.

4) Please provide additional data investigating the importance of other residues (E168, R371, Y330 and A171, M157 and L391) in the S-pocket to bolster the data with R209.

To investigate the importance of S-pocket in PG-moiety binding, additional binding studies of N-Acetylmuramyl-L-alanyl-D-isoglutamine have been performed with S-pocket mutants E207A and Y330F to bolster data with R209E mutant.

– New Figure 1B shows a ThermoFluor data with change in melting temperature (∆Tm) verses increasing concentration of PG-moiety. Data represents PG-moiety binding studies with different S-pocket mutants (R209E, E207A, Y330F).

– New Figure 1C contains ThermoFluor data on differential fluorescence change (-dF/dT) verses PG-moiety concentrations. A new Figure 1 – source data 1 supports Figure 1C.

Reviewer #1 (Recommendations for the authors):The structures of LdtMt2 have been published before, both as apo and in complex with a β-lactam molecule. The current structures report on a novel allosteric site that was not previously described.Figure 2A should be moved to Figure 1.

We appreciate the suggestion from the reviewer on moving Figure 2A to Figure 1. However, Figure 2A supports the experimental data in Figure 2B. Both figure 2A and 2B can make it easy for the readers to review several domains in Ldt_Mt2_ and correlate with experimental results on ß-lactam hydrolysis activity. We have also put one additional Figure 2C on Circular dichroism (CD) spectra of R209E mutant and Wild-type to rule out any major secondary structural changes that could impact the ß-lactam hydrolysis activity of the enzyme.

– New Figure 2B contains ß-lactam hydrolysis data for all truncated domains and mutations in S-pocket into one single figure. A new Figure 2 – source data 1 supports Figure 2B.

– New Figure 2C contains CD spectra of wild-type and R209E mutant.

The structures in Figure 1C (purple and green) that are superimposed do not look superimposed. No quantitative measure (RMSD) is provided. What parts of the structure were superimposed? Perhaps a sequence alignment between LdtBS and LdtMt2 might explain this better. Further qualitative analysis of the inner cavity should be carried out. The electrostatic surface should be shown for both proteins and how the corresponding electrostatic surface of the PG complements it should be illustrated.

With suggestion from the reviewer #1, we again superposed the catalytic domain of Ldt_BS_ with YkuD domain of Ldt_Mt2_ and these domains superposed with an RMSD of 1.46 Å. Ldt_BS_ has a LysM domain that is missing in the Ldt_Mt2_ structure. In the Ldt_BS_ structure, the acidic sugar moieties of PG chain bind in-between the LsyM domain and catalytic domain across a positively charged groove. Based on the structural details of PG binding in Ldt_BS_ (Schanda *et al.*, 2014) and Ldt_Mt2_ (Figure 1A), a PG chain was computationally placed over a positively charged surface across the IgD2-YkuD domain interface encompassing the S-pocket. The positively charged electrostatic surface of Ldt_BS_ and Ldt_Mt2_ have been illustrated that binds PG chain.

– New Figure 1D shows superposition of catalytic domains of Ldt_Mt2_ (green) and Ldt_BS_ (pink)_._

– New Figure 1E shows the structures of Ldt_BS_ and Ldt_Mt2_ with their surface charge representation and with proper labeling of their domains to highlight the structural differences between them.

Dynamic distances between important residues (H336-C354, H336-S351, H336-S337, etc.) over the course of the simulation should be plotted to support the claims that the authors make.

An additional work on MD simulation at 200 ns has been performed. Network analysis has also been performed to search allosteric changes in Ldt_Mt2_ that could emanate from the S-pocket (Figure 3, and Figure 3—figure supplement 1).

Addressed in response to essential revision# 1

One of the strengths of this paper is the finding that S351 and not S337 might form the catalytic triad. The authors should strengthen this by quantifying structural interactions that support mutagenesis experiments. A wealth of data is present in the MD simulations; they should use this to their advantage.

A new data on network analysis of MD at 200 ns has been added to further understand the role of S351 and its modulation by the allosteric network (Figure 3B and Figure 3—figure supplement 1).

Our experimental and MD results at 200 ns followed by network analysis have clearly demonstrated new findings on the role of S351 in the catalysis that can be influenced by S-pocket (Figure 3 and Figure 3—figure supplement 2). However, new investigations with 200 ns MD data suggest that S337 may not be replaced by S351 in the catalytic triad. Instead, S351 residue is a new residue we have identified that can stabilize the H336-NE1 through hydrogen bond interactions. We have changed our statement in the main-text about S351 role in the catalytic triad. We highlight the important role of S351 in the process of catalysis through interactions with H336-NE1 that can be modulated by allostery. Based on our current level of computational results, we do not think the replacement of S337 by S351 in the catalytic triad, rather, both of these residues seem to be important for the catalytic center, A deeper computational study will be required to investigate the roles of S337 and S351 in catalytic process. However, our results clearly demonstrate the role of S351 in interaction with H336 in the catalytic process and this information is very new and pave the way for more independent investigational studies in future.

Addressed in response to essential revision# 1

Since there are other structures of mutants of LdtMt2 present, why did the authors not try to crystallize R290E and other mutants? The structural findings would have been definitive in assessing the structural differences between the apo and ligand-bound states. A comment should be made in the discussion.

We had tried several attempts to crystallize R209E mutants but were unsuccessful.

Based on the new data on MD simulations and network analysis and the crystal structure with an experimental carbapenem drug T203, we have provided strong evidence on the existence of allosteric propagation between the S-pocket and catalytic site that regulates enzyme function (Figure 3 and Figure 3—figure supplement 1) and Figure 6D. MD data and the high-resolution crystal structure data are matching on allosteric propagation between S-pocket and catalytic site.

On the advice of the reviewer#, we have added a new comment in the discussion:

“This study provides a high-resolution crystal structure of Ldt_Mt2_ with an experimental carbapenem T203 that can be the basis for understanding structure-activity relationship (SAR) data on ß-lactam binding in the S-pocket and catalytic site. Insights gained from this analysis may guide the acquisition and/or the design and synthesis of new inhibitors. Additional structural and computational studies of the S-pocket and catalytic mutants will be required to gain a more detailed understanding of the catalytic process and ß-lactam binding that can be modulated by the S-pocket”.

The manuscript relies heavily on MD simulations to structurally explain a number of experimental observations. The computational study is an essential part of the manuscript, yet it seems to be carried out like an afterthought. MD simulations analysis is poorly carried out. The details in the method section for both automated docking and MD simulations are poor and lack details to reproduce the study. For example, How were the ligands parameterized? No evidence has been provided which even remotely suggests that the simulated system has been equilibrated, let alone converged. How many simulations were run to confirm the observations were statistically significant? The interpretations of the results are equally perplexing. The tumbling of the ligand in the binding site is because the system has not equilibrated and should be omitted from being included in any interpretation (line 334-335). 100ns might be too short for the essential interactions to stabilize. If the authors think otherwise, they should support that with evidence. None of that has been included. MD simulation figures in the supplementary are poor and do not provide any information that strengthens the excellent biochemical analysis and mutagenesis experiments. The MD simulation figures in Supplementary are not even labeled appropriately.

Based on the advice, line 334-335 has been edited to avoid over-prediction with the current level of MD data on biapenem binding.

We have provided additional data as an alternative to represent S-pocket binding. Our experimental results with additional mutations in the catalytic site and S-pocket bolster the data on PG-substrate and ß-lactam binding in S-pocket. This additional mutational data clearly demonstrates the relationship between S-pocket and catalytic site during ß-lactam binding (Figure 1B and 1C; Figure 4C and D).

We have also provided a more detailed network analysis of MD results that matches with the results of crystal structure data on allosteric propagation (Figure 3B and Figure 3—figure supplement 1)

(Kindly see response to essential revision# 3 and essential revision# 4).

We have used MD simulation to further understand and illustrate biapenem binding in both S-pocket and catalytic site (Figure 4E). What have tied to convey through the MD simulations whether the binding of biapenem in S-pocket alone and/or the catalytic site would show any level of cooperativity to support our experimental results. A drug alone in the S-pocket or catalytic pocket is not able to stay for long and moves out within 40ns. However, simultaneous ß-lactam drug binding in both the S-pocket and catalytic pocket certainly delays the exit of biapenem from the S-pocket and stabilizes the drug in catalytic site. We have properly labelled the MD time in figure 4E to understand biapenem binding with MD simulation time. We have also properly labeled Figure 4—figure supplement 1; Figure 4—figure supplement 2 and Figure 4—figure supplement 3.

Why have the dynamics of R209 not been discussed? If the allosteric signals are indeed emanating from this residue (or any in the allosteric pocket), then surely a direct link could be traced between R209 (or other residues) and the catalytic triad? A network analysis should be able to resolve this. Or some equivalent method that demonstrates this structural link.

With advice from the reviewer, we have performed dynamic network analysis to identify allosteric changes that propagate between S-pocket and catalytic site (Figure 3B and Figure 3—figure supplement 1). We find that these allosteric changes that were computed in the network analysis of MD data match with the allosteric changes that were observed in the crystal structure (Figure 6D).

We have added new results on network analysis in the manuscript.

Line 342/344/346: There is no Figure S4A or S4B. Just S4

This was a typo. They were actually new Figure 4—figure supplement 3A and Figure 4—figure supplement 3B

Line 347-350: "Thus, MD simulations suggest that biapenem binding across the S-pocket surface imposes stability in fluctuations of β-lactam movement in the catalytic site and the.-lactam ring carbonyl group maintains a close distance with the Sϒ atom of C354 that favor a nucleophilic attack"It might very well be that this is indeed the case. But without evidence from MD simulation, it seems like they are an over-interpretation of results.

Based on the advice, line 347-350 has been edited to avoid over-prediction with the current level of MD data on biapenem binding.

The docking protocol was baffling. When the authors have a crystal structure with a ligand, then what was the reason they did not carry out a template-based docking? Surely, the β-lactam ring in T203 could have been used as a fixed focal point. Yet, they even dock T203. If the authors were trying to do a blind dock to confirm the binding mode of T203 (because of the poor electron density), then they should say so and also show the results of the dock, superimposed on the crystal structure. No such thing has been detailed in the method or discussion.

Blind docking around the catalytic site (within 60 Å radius) was performed to rule out any biasness in docking results influenced by the experimental data. We have added the docking details in the Materials and methods.

Kindly see response to essential revision# 3.

The crystal structure of Ldt_Mt2_ in complex with experimental carbapenem T203 was solved later after discovering the S-pocket and all docking studies with ß-lactams. We have added a new figure (Figure 6—figure supplement 1E) on the superposition of T203 drug from docking result (green colour) with T203 drug from crystal structure (pink colour).

The authors mention that the electron density of T203 is poor. Therefore in addition to 2Fo-Fc maps, they should also include omit maps. This will only strengthen their claim of how they have modeled the ligand in the sites.

Fo-Fc omit map in both S-pocket and catalytic pocket have been calculated and are shown in new Figure 6—figure supplement 1A and 1B.

Line 449: What was this distance measured between? Which residues or atoms?

Distance of catalytic residue C354 with several residues in the S-pocket was measured (R209E, P169, A171 and others) and an average came to ~21 Å distance.

Line 452-453: "Mutational changes in the S-pocket or the catalytic pocket nullifies all the allosteric communications that are otherwise important in the cooperative binding of dual ß-lactams." How is this structurally achieved between the two sites that are far apart? The authors should use network analysis to resolve this?

A new data on network analysis (kindly see new Figure 3 and Figure3—figure supplement 1) and crystal structure with T203 (Figure 6) highlight the details of allosteric propagation and changes in the catalytic center.

Kindly see response to essential comment#1

Line 471-472: There is surely a mechanism of allosteric communication, its just that the authors have not provided evidence for that. What the authors have shown is the end states in either the allosteric site or the catalytic site, but not how signals would be communicated upon ligand binding in the allosteric site to the catalytic site.

We have provided new evidence on the network of allosteric propagation in ldt_Mt2_ by analyzing MD data at 200 ns (Figure 3B and Figure 3—figure supplement 1). These results match with the allosteric changes observed in the crystal structure with T203 drug (Figure 6D).

We have added a new paragraph on network analysis in the result section.

Also, in the discussion, we have added a new line

“Additional structural and computational studies of the S-pocket and catalytic mutants will be required to gain a more detailed understanding of the catalytic process and ß-lactam binding that can be modulated by the S-pocket”.

Line 479: Why have the properties (structural or sequence) have not been shown for the putative S-pocket. A close-up (structural superimposition of conserved residues) or sequence alignment should have been presented. Perhaps figures of electrostatic charge surface would convey the point the authors are trying to make. Figure 8 is the least informative as it currently stands.

A new figure, Figure 7—figure supplement 1F, has been added. A closeup of S-pocket residues are shown for different LDTs in *Mycobacterium tuberculosis*.

Reviewer #2 (Recommendations for the authors):1. After modeling glucose into the S-pocket, authors propose R209, E168, R371, Y330 and A171, M157 and L391 as residues interacting with the sugar. However only a single mutation (R209E) disrupts binding of the PG precursor. The identification of only a single mutation is strange, because for a bonafide binding site, there should be at least > 1 residue which can disrupt binding. Because other mutations are not reported that disrupt binding at S-pocket, there is not strong evidence that the S-pocket is an actual site for binding ligands. This poses an important question: is it possible that R209E affects Ldt-Mt2 function by just disrupting the IgD2-YkuD domain interface and subsequently destabilizes the active site? To confirm that the S-pocket is actually a distinct site that binds ligand and allosterically regulates the active site, there needs to be evidence that R209E does not merely disrupt dimerization of the two domains. The majority of the case for the existence of this pocket hinges on functional studies with R209E, therefore this possibility should be ruled out.

Following the suggestions from reviewers#2, we have performed additional mutations in the S-pocket to bolster the data on PG moiety.

Addressed in response to essential revision# 4.

In addition, to rule out any effect of R209E mutation on the secondary structure of Ldt_Mt2_, we performed CD spectra. CD spectra did not suspect any alteration in the secondary structure of the enzyme upon R209E mutation in the S-pocket. (Kindly see new Figure 2C).

2. Authors have extracted quite a bit of mechanistic detail from their simulations, including demonstrating cooperativity between the two sites. However authors should be cautious, because 100 ns trajectories are not long enough to give meaningful information about fluctuations or persistence of conformations. For a protein with 450 residues, my sense is that the trajectories are barely equilibrated at 100 ns. However RMSDs over the trajectory can be shown to give some indication of how overall structure is changing between 0 and 100 ns. This will help the authors determine whether the changes in active site residues that they report can be expected to persist over longer timescales. One suggestion to achieve longer trajectories: authors could truncate the complex to just the two domains of interest, which might allow increased simulation time to further support these claims.

In full agreement with reviewer#2, we have performed additional MD simulation at 200 ns with 2 domains only (IgD2-YkuD).

A new paragraph has been added on network analysis in the Results section.

Addressed in response to essential revision# 1.

This is a potentially exciting piece of work. To make this suitable for publication, authors must provide a little more evidence that the S-pocket is a site that binds ligand. For both ligands modeled into electron density, the observation of only small fragments of these ligands casts doubt on the veracity of their binding. Likewise, the identification of only a single mutation that disrupts S-pocket is not sufficient to support binding.Additionally, for some of the claims being made using docking and MD simulations, particularly those of cooperativity and pocket cross-talk, authors have not demonstrated that their trajectories are long enough to extract these conclusions. Evidence should be provided to test equilibration and confirm that the key interactions/conformational changes presented here will persist in longer simulations. Alternatively, authors could add additional experimental validations to demonstrate cooperativity between the two sites.

We have performed additional MD at 200 ns followed by network analysis to discovery allosteric propagation. These results match with the allosteric changes observed in the high-resolution crystal structure with an experimental carbapenem T203. (Figure 3 and Figure 3—figure supplement 1).

As an alternative, we have performed additional mutations in the S-pocket and catalytic center to further bolster claim in Biapenem binding in S-pocket and catalytic site in cooperativity (Figure 1 and Figure 4).

Addressed in response to essential revision# 1, essential revision# 3 and essential revision# 4

Reviewer #3 (Recommendations for the authors):Abstract: the first sentence suggests that only M. tuberculosis has Ld transpeptidases, which is incorrect. Please edit this accordingly. The site termed 'S-pocket' on Line 42 is referred to as 'S-site' on line 45; please adopt a consistent terminology.

With the suggestion from reviewer#3, first line of the abstract has been changed to “L,D-transpeptidase function predominates in atypical 3 – 3 transpeptide networking of peptidoglycan (PG) layer *in Mycobacterium tuberculosis*.”

Also, S-pocket has been referred consistently.

Introduction: although it is well-written, I do not think the summary of the paper (Lines 149-155) accurately reflect the results of the paper. In my view, the results on β-lactams and not peptidoglycan are the significant portion of the paper. Also, claims such as 'several interdisciplinary approaches' and 'explain the mechanism for the manifestation of activity of Ldtm2' are not completely supported, since the authors uncover an additional facet of the functioning of the enzyme. These sentences should be tempered.

We are highly grateful to the reviewers#3 for these suggestions. We have made changes in the last sentence of introduction.

Line 161: it is not clear whether the crystal structure at 1.57A is an improvement on previous efforts and whether this improved resolution is what enables the authors to discover the glucose binding in the S-pocket. The text should be re-written to make these points clear.

As per the advice from reviewer#3, a new sentence has been added

“This high-resolution crystal structure was an improvement of our previous efforts (Erdemli *et al.*, 2012; Kumar *et al.*, 2017) that enabled us to identify an electron density in a pocket”

Figure 1B: comparable data for glucose would help the readers assess the strength of the conclusions regarding PG. In this panel, and in similar panels in other figures, it is not clear if the trend is significant. Could the authors show error bars for such figures, and perhaps include a plot of Tm vs. substrate concentration in the inset to highlight the trend better?

With suggestions from reviewer#3, a ∆Tm vs substrate concentration plot has been placed in new Figure 1B**.**

Additional mutations have been done in the S-pocket to bolster the data on PG-sugar moiety binding with S-pocket.

Kindly see response to essential revision# 4.

Figure 1D: the inner and outer cavity in the catalytic site that is discussed in the text is not labelled in the figure, and so it is hard to appreciate the location of the larger pentameric PG molecule. Furthermore, the authors could show Thermofluor data for a suitable point-mutant in the inner cavity to confirm their claims that the PG molecule can be located within the S-pocket and the inner cavity and complement the data in Figure 1B.

With suggestions from reviewer#3, inner and outer pocket of catalytic have been properly labeled in Figure 1E.

In our understanding, mutation in the inner pocket would not affect the binding of PG-sugar moiety at S-pocket. Only the stem peptide of PG chain would bind the inner pocket. We lack a full native PG substrate that could bind both S-pocket and inner pocket in one go. That is why we choose Biapenem for our study that could bind both S-pocket and catalytic site.

We have a line in the discussion “prior to 3-3 transpeptide linkage, both S-pocket and catalytic site may work in cooperativity to facilitate synchronous binding of two PG substrates (donor and acceptor substrates); however, this requires experimental validation using nascent PG substrates that are beyond the scope of our study**”.**

As we lack a native substate for studying the catalysis and allostery, that is why we choose ß-lactam for observing the catalytic activity.

The location of inner peptide has been reported in earlier studies and this pocket is supposed to bind the acceptor stem peptide of PG (Erdemli et al., 2012). Structure of PG with Ldt_BS_ further supports our modeling study for PG stem binding in inner pocket.

Lines 200-205: it is unclear what the phrases 'deletion of IgD2 from the YkUD domain as assessed from the YkuD domain alone' means. At Line 205, the S-pocket is described as being carried by the IgD2 domain but

Based on the point noticed by the reviewer, we have made a small change in the sentence

“This suggests an important role of S-pocket which is partly carried by the IgD2 domain in governing the catalytic activity of Ldt_Mt2_ enzyme”.

Lines 271-273: while I do not disagree qualitatively with the descriptions, there is clearly a large shift in Tm at the lowest concentration of bioapenem, which is not reflected in the text.

In response to reviewer#3, we have added new data on biapenem binding with Ldt_Mt2_ and mutants that clearly reflects a saturable binding in the S-pocket. A new Figure 4D has been added.

Kindly see response to essential revision# 3.

Line 304: in contrast, the decreased stability of the S351A mutant has nothing to do with the biapenem binding – the statement describing this is misleading, Please revise this.

With reviewer’s advice, the sentence has been removed.

Figure 4B: based on the statements in the text, the effect of biapenem binding on melting structure is completely abroagated by the R209E and S351A mutations, yet the graphs show red arrows for the change in Tm. Please remove these if the trend is non-significant.

T_m_ arrows have been removed from S351, R209E and other mutants.

Figure 4C: given the content-rich format of eLife, the authors should provide animations of the simulations they describe, which would be far more informative than the snapshots presented.Figure presentation: in many figures, the red arrow labels are hard to see. Please make the arrows bigger.

Arrows have been marked bigger in new Figure 4E (this corresponds to earlier Figure 4C). Figures have been properly labeled properly.

Figure 6 and last section of the Results: the motivation for studying in detail the molecule T203 is unclear. Is it because it has the most dramatic effects based on the results of Figure 5? The authors should cite relevant publications for this drug and include in the discussion how such a finding on an experimental carbapenem may be beneficial, as this is a strength of this manuscript.

We have added the reference for experimental carbapenem T203 in the results, (Pankaj Kumar et al., *Nature Chemical Biology*, 2017).

With suggestion from reviewer, we have added a new sentence in the discussion

“This study provides a high-resolution crystal structure of Ldt_Mt2_ with an experimental carbapenem T203 that can be the basis for understanding structure-activity relationship (SAR) data on ß-lactam binding in the S-pocket and catalytic site. Insights gained from this analysis may guide the acquisition and/or the design and synthesis of new inhibitors. Additional structural and computational studies of the S-pocket and catalytic mutants will be required to gain a more detailed understanding of the catalytic process and ß-lactam binding that can be modulated by the S-pocket.”

Figures and figure legends: many methodological details e.g., the software used for the curve fitting can be moved to the Materials and methods and do not need to be repeated for every figure.

The changes have been made accordingly.

[Editors’ note: further revisions were suggested prior to acceptance, as described below.]

The manuscript has been improved but there are some remaining issues that need to be addressed, as outlined below:1) S-pocket: R209 continues to be the only residue with a significant impact on ligand binding/catalysis. Y330 was used as a second proxy for the S-site, but its effect is much more muted than R209 or E207. The data with E207A, while not comprehensive (e.g. nitrocefin hydrolysis), does support the S-site more than data with Y330F – please note this.

Nitrocefin hydrolysis assay has also been performed with E207A mutant to compare with wild-type. We have provided new Figure 2 —figure supplement 1 to support Figure 2.

2) The CD spectra of wt vs R209A show some differences ~ 210 nm that could be explained by reduced β-sheet content in the spectrum of the variant protein. Such a change could indicate problems with the protein fold that might account for the differences in catalytic activity. Can the spectra be fit to determine whether this difference is meaningful? This will bolster the data suggesting that R209 forms part of an allosteric site.

We have added a new explanation about the CD spectra data in the manuscript. Several of the experiments in the manuscript, including: (a) CD spectra, (b) ThermoFluor and (c) Molecular dynamic simulations suggest no major change in overall folding of the protein due to R209E mutation.

3) Cooperativity between the catalytic site and the active site: there is no clear evidence that cabapenam binding occurs cooperatively. In the absence of a quantitative assessment of cooperativity, it might be best to state that this is a possibility rather than a phenomenon that has been determined.

With the suggestions from editor, we have made necessary changes in the statement on cooperativity. However, with a new data on MD simulations of the Ldt_Mt2_-T203 crystal structure (Video 1, Video 2 and Figure 6 —figure supplement 2 are provided), a cooperativity in dual ß-lactam binding is clearly evident that has been explain in the last paragraph of the Results.

4) The binding of T203 to the active site is clear, but the 2Fo-Fc map shows low occupancy at the S-site. Were greater concentrations of the drug used? IF so, do those data reveal improved occupancy at the S-site?

In our study with crystallization of Ldt_Mt2_ with carbapenems, including T203, we have gone up to 6mM concentration, but could never improve the occupancy in S-pocket. However, our MD simulation data with Ldt_Mt2_-T203 clearly explains that binding of T203 in the S-pocket is influenced by the binding in catalytic pocket (Video 1, Video 2 and Figure 6 —figure supplement 2). T203 drug does not bind alone in the S-pocket.

Additionally, the comments below were provided by a reviewer.Reviewer #2 (Recommendations for the authors):Ahmad et al., have addressed my largest concern, going to extra steps to make sure that other S-pocket mutations disrupt small molecule binding. This portion of the manuscript is more convincing and the discovery of the S-pocket is quite novel.While simulations have been increased, the authors still did not address the question of equilibration or show RMSDs over the length of the trajectory to show the feasibility of extracting mechanistic details from these simulations. However, the description of the conformational differences between wildtype and mutant Ldt-Mt2 is quite compelling.Overall, concerns still remain about the computational work.1. Authors mention that they have computed a dynamic network analysis for their MD trajectories. There is no explanation of what this analysis means or what information it provides. The method section is not instructive about the nature of this analysis, except to say that it was performed using VMD. It is also not my understanding that this method is used to calculate 'distances', as these can be calculated directly from the trajectory. To my knowledge, dynamic network analyses should minimally include discussions of one or more of the following: nodes and/or edges, suboptimal communication pathways, community structures, correlations, etc. If this detail about dynamic network analysis remains, there needs to be an explanation of what the approach is (explicit details in the Methods section), what information was generated for their trajectories and if possible, figures showing exactly what was extracted. As it stands now, all that is present are these distance plots and the density representations, which are informative but not dynamic networks. See https://doi.org/10.1073/pnas.0810961106 as a reference.

We are so obliged to the reviewer for his continuous support in improving our manuscript by his critical comments.

Based on the suggestions from reviewer, network analysis has been done to identify the pathways of allosteric communications between S-pocket and catalytic site. Kindly find new Figure 3 —figure supplement 3 and Figure 3 – source data 3.

2. The docking study with biapenem is a weakness of this work. Because this part relies on docked and not crystal structures, the premise is tenuous to begin with. The fact that the small molecules fly out by 40 ns even in the best case is not reassuring that there is real binding. The most convincing way to use MD to make this point would be to use the crystal structure with the T203 at both sites as a starting point. My strong recommendation would be to remove this portion of the study or strengthen the work significantly by repeating the study with T203 and seeing whether the previously observed cooperativity persists.

With suggestions from the reviewer, following changes have been made in the manuscript:

a. Additional details of MD simulation with biapenem have been added in the method sections.

b. MD simulation has also been performed with LdtMt2-203 crystal structure to identify the cooperativity in dual ß-lactam binding. Kindly find (Video 1, Video 2 and Figure 6 —figure supplement 2).